# ORTHOGONAL ESTIMATION OF DIFFERENCE OF $Q$-FUNCTIONS

## ABSTRACT

Offline reinforcement learning is important in many settings with available observational data but the inability to deploy new policies online due to safety, cost, and other concerns. Many recent advances in causal inference and machine learning target estimation of "causal contrast" functions such as CATE, which is sufficient for optimizing decisions and can adapt to potentially smoother structure. We develop a dynamic generalization of the R-learner (Nie et al., 2021; Lewis & Syrgkanis, 2021) for estimating and optimizing the difference of $Q^\pi$-functions, $Q^\pi(s, a) - Q^\pi(s, a_0)$, for potential discrete-valued actions $a, a_0$, which can be used to optimize multiple-valued actions without loss of generality. We leverage orthogonal estimation to improve convergence rates, even if $Q$ and behavior policy (so-called nuisance functions) converge at slower rates and prove consistency of policy optimization under a margin condition. The method can leverage black-box estimators of the $Q$-function and behavior policy to target estimation of a more structured $Q$-function contrast, and comprises of simple squared-loss minimization.

## 1 INTRODUCTION AND RELATED WORK

Learning optimal dynamic treatment rules, or sequential policies for taking actions, is important, although often only observational data is available. Many recent works in offline reinforcement learning develop methodology to evaluate and optimize sequential decision rules, without the ability to conduct online exploration.

Offline reinforcement learning is closely connected to causal inference. An extensive literature on causal inference and machine learning establishes methodologies for learning *causal contrasts,* such as the *conditional average treatment effect* (CATE) (Wager & Athey, 2018; Foster & Syrgkanis, 2019; Künzel et al., 2019; Kennedy, 2020), the covariate-conditional difference in outcomes under treatment and control, which is sufficient for making optimal decisions. A key "inductive bias" motivation is that the causal contrast (i.e. the difference that actions make on outcomes) may be smoother or more structured (e.g., sparser) than what the main effects (what happens under either action by itself). Therefore methods that specifically estimate these contrast functions could potentially adapt to this favorable structure when it is available. A classically-grounded and rapidly growing line of work on double, orthogonal, or debiased machine learning (Kennedy, 2022; Chernozhukov et al., 2018) derives improved estimation procedures for these targets. Estimating the causal contrast can be statistically favorable, while sufficient for decision-making.

In this work, building on recent advances in heterogeneous treatment effect estimation, we focus on estimating analogous causal contrasts for offline reinforcement learning, namely $\tau_t^\pi(s; a, a_0) = Q_t^\pi(s, a) - Q_t^\pi(s, a_0)$, for possible actions $a, a_0$ in the action space $\mathcal{A}$. We focus on the case of two actions, though the method generalizes to multiple actions. This is closely related to, but different from advantage functions in reinforcement learning, defined as $Q^\pi(s, a) - V^\pi(s)$, the advantage of taking action $a$ beyond the policy. For brevity, to describe statistical behavior, we denote the binary-action difference-of-$Q$ $\tau_t^\pi(s) = Q_t^\pi(s, 1) - Q_t^\pi(s, 0)$, with the understanding that our analysis in the sequel is a tool for the multiple-action setting as well.

The sequential setting offers even more motivation to target estimation of the contrast: additional structure can arise from sparsity patterns induced by the joint (in)dependence of rewards and transition dynamics on the (decompositions of) the state variable. A number of recent works point out this additional structure (Wang et al.; 2022), for example of a certain transition-reward factorization, first studied by (Dieterich et al., 2018), that admits a sparse $Q$-function contrast (Pan & Schölkopf, 2023). (Zhou, 2024b) proposes a variant of the underlying blockwise pattern that also admits sparse optimal $Q$ functions and policies. Section 1 illustrates how both of these structures have very different conditional independence assumptions. Methods designed assuming one model is correct may not perform well if it is not. However, both structures imply the same property on the difference-of-$Q$ functions: sparsity in the endogenous

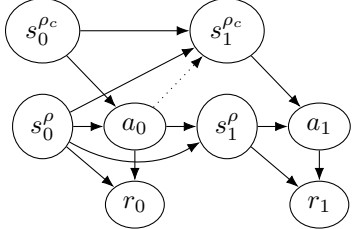
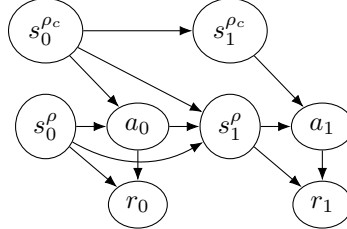

Figure 1: (Left) Reward-relevant/irrelevant factored dynamics of (Zhou, 2024b). (Right) Exogenous-Endogenous MDP model of (Dietterich et al., 2018). The dotted line from $a_t$ to $s_{t+1}^{\rho_c}$ indicates the presence or absence is permitted in the model.

component. Our method can adapt to such underlying sparsity structure when it is present in the $Q$-function contrast, in addition to other scenarios where the contrast is smoother than the $Q$-functions themselves.

The contributions of this work are as follows: We develop a dynamic generalization of the R-learner (Nie & Wager, 2021; Lewis & Syrgkanis, 2021) for estimating the $Q$-function contrast. We show theoretical guarantees of improved convergence rates. Therefore our method leverages behavior policy estimation to improve estimation without suffering from unstable propensity weights. We illustrate benefits of adapting to structure in synthetic examples.

**Related work for our approach.** There is a large body of work on offline policy evaluation and optimization in offline reinforcement learning (Jin et al., 2021; Xie et al., 2021), including approaches that leverage importance sampling or introduce marginalized versions (Jiang & Li, 2016; Thomas et al., 2015; Kallus & Uehara, 2019a; Liu et al., 2018). For Markov decision processes, other papers study statistically semiparametrically efficient or doubly-robust estimation, but of the *averaged policy value* $\mathbb{E}[V_1^{\pi^e}(S_1)]$, rather than MSE convergence of the difference-of-Q function as we do here (Kallus & Uehara, 2019a;b; Xie et al., 2023; Kallus & Zhou, 2022). The literature on dynamic treatment regimes (DTRs) studies a method called advantage learning (Schulte et al., 2014), although DTRs in general lack reward at every timestep, whereas we are particularly motivated on sparsity implications that arise jointly from reward and transition structure. In particular, beyond policy value estimation, we aim to recover the entire contrast function.

Advantage functions are widely studied in RL and dynamic treatment regimes (Neumann & Peters, 2008; mur, 2003). However, policy optimization is hard because *which* contrast it evaluates is time-$t$ policy dependent when optimizing at time $t$. Our difference-of-$Q$ functions are independent of candidate time-$t$ policies when optimizing at time $t$.

(Pan & Schölkopf, 2023) note the analogy with causal contrast estimation and derive a Q-function independent estimator, but in the online setting. After preparing an initial version of this paper, we became aware of the recent work of Pan & Schölkopf (2024) in the offline setting. While their method elegantly avoids estimating future $Q$ functions, it requires a nonconvex constraint on the action-average of advantages, which is computationally and statistically difficult. The motivations are different; the methods are complementary. (See appendix for more discussion.) Our identification argument is different and we focus on improved statistical guarantees.

At a high level, our method is similar to the dynamic DR learner studied in (Lewis & Syrgkanis, 2021) in that we extend the R-learner identification approach to a sequential setting, although the estimand is quite different. In particular, generalize structural nested-mean models (SNMMs) by estimating "blip-to-zero" functions but only consider heterogeneity based on a fixed *initial* state and dynamic treatment regimes with terminal rewards.Consequently, our analysis is similar at a high level. Overall, a closely related work with a similar goal of estimating contrast functionals in reinforcement learning is that of (Shi et al., 2022), which derives a pseudo-outcome for estimating the $Q$-function contrast in the infinite horizon setting. We focus on the finite-horizon setting and have a different estimation strategy building on the $R$-learner Note that the (single-stage) R-learner loss function is an overlap-weighted (Li et al., 2019) regression against the doubly-robust score (DR-learner (Kennedy, 2020)). (See (Morzywolek et al., 2023; Chernozhukov et al., 2024)).

We briefly discuss other approaches. (Nie et al., 2021) studies OPE for advantage functions in a special case of optimal stopping. Prior works that consider policy optimization under a restricted function class introduce difficult policy-dependent nuisance functions. Our policy optimization does not restrict functional complexity, which would otherwise require approximation or re-estimating nuisance functions every *iteration* of policy parameter optimization,

as in (Xu et al., 2021). We do make a margin assumption to relate convergence of Q-function contrasts to policy value convergence, analogous to (Shi et al., 2022).

## 2 METHOD

**Problem Setup:** We consider a finite-horizon Markov Decision Process on the full-information state space comprised of a tuple $\mathcal{M} = (\mathcal{S}, \mathcal{A}, r, P, \gamma, T)$ of states, actions, reward function $r(s, a)$, transition probability matrix $P$, discount factor $\gamma < 1$, and time horizon of $T$ steps, where $t = 1, \ldots, T$. We let the state spaces $\mathcal{S} \subseteq \mathbb{R}^d$ be continuous, and assume the action space $\mathcal{A}$ is finite. A policy $\pi : \mathcal{S} \mapsto \Delta(\mathcal{A})$ maps from the state space to a distribution over actions, where $\Delta(\cdot)$ is the set of distributions over $(\cdot)$, and $\pi(a \mid s)$ is the probability of taking action $a$ in state $s$. (At times we overload notation a bit so that $\pi(s) \in \mathcal{A}$ indicates the action random variable under $\pi$ evaluated at state $s$). Capital letters denote random variables $(S_t, A_t, \dots)$, lower case letters $s, a$ denote evaluation at a generic value.

The reward is a function of current state and action, hence a random variable; denoted as $R_t = R_t(S_t, A_t)$. The value function is $V_t^\pi(s) = \mathbb{E}_\pi[\sum_{t'=t}^T \gamma^{t'-t} R_{t'} \mid s]$, where $\mathbb{E}_\pi$ denotes expectation under the joint distribution induced by the MDP $\mathcal{M}$ running policy $\pi$. The state-action value function, or $Q$ function is $Q_t^\pi(s, a) = \mathbb{E}_\pi[\sum_{t'=t}^T \gamma^{t'-t} R_{t'} \mid s, a]$. These satisfy the Bellman operator, e.g. $Q_t^\pi(s, a) = r(s, a) + \gamma \mathbb{E}[V_{t+1}^\pi(s_{t+1}) \mid s, a]$. The optimal value and q-functions are denoted $V^*, Q^*$ under the optimal policy. We focus on estimating the difference of $Q$-functions (each under the same policy), $\tau_t^\pi(s) = Q_t^\pi(s, 1) - Q_t^\pi(s, 0)$. We focus on the offline reinforcement learning setting where we have access to a dataset of $n$ offline trajectories, $\mathcal{D} = \{(S_t^i, A_t^i, R_t^i, S_{t+1}^i)_{t=1}^T\}_{i=1}^n$, where actions were sampled under a behavior policy $\pi^b$. We state some notational conventions. For some generic function $f$, and a choice of norm $u \geq 0$, we define the $u$-norm $\|f(X)\|_u := \mathbb{E}[\int |f(x)|_u \cdot p(x)]^{1/u}$. In the context of estimation (rather than discussing identification), we denote the true population functions with a $\circ$ superscript, i.e. $\tau_t^{\pi, \circ}$ and so on.

**Policy evaluation: Identification.** First we overview deriving the estimating moments of our approach. The arguments are broadly a generalization of the so-called residualized R-learner (Nie & Wager, 2021); (Lewis & Syrgkanis, 2021) considers a similar generalization for structural nested mean models without state-dependent heterogeneity. For this section, we discuss the true population $Q, \tau$, etc., functions without decoration with $\circ$, introduced later in the context of estimation.

Our main identifying assumption is that of Markovian sequential unconfoundedness.

**Assumption 1** (Sequential unconfoundedness). *The underlying dynamics are from a Markov Decision Process $\mathcal{M}$, and under the behavior policy $\pi_b$, actions $A_t$ were taken with probability depending on observed states $S_t$ alone (and not on any unobserved states).*

This assumption posits that the state space is sufficient for identification. Informally, it assumes the behavior policy was acting on observed states alone, rather than in a POMDP. Assumption 1 is basically the implicit assumption in RL based on MDP rather than POMDP. Denote $\underline{\pi}_{t+1} = \pi_{t+1:T} := \{\pi_{t+1}, \ldots, \pi_T\}$ for brevity. Then $Q_t^{\underline{\pi}_{t+1}}$ indicates the $Q_t$ function under policy $\pi$. For brevity, we further abbreviate $Q_t^\pi := Q_t^{\underline{\pi}_{t+1}}$ when this is unambiguous. We seek to estimate:

$$\tau_t^\pi(S_t) = Q_t^\pi(S_t, 1) - Q_t^\pi(S_t, 0), \tag{1}$$

Note that the $Q$-function satisfies: $Q_t^\pi(S_t, A_t) = \mathbb{E}[R_t + \gamma Q_{t+1}^\pi(S_{t+1}, A_{t+1}) \mid S_t, A_t]$. Under sequential unconfoundedness and Markovian properties, we obtain the conditional moment:

$$\epsilon_t = R_t + \gamma Q_{t+1}^\pi(S_{t+1}, A_{t+1}) - \{Q_t^\pi(S_t, 0) + A_t \tau_t^\pi(S_t)\}, \text{ and } \mathbb{E}[\epsilon_t \mid S_t, A_t] = 0. \tag{2}$$

Define the analogue to the marginal outcome function, which is the state-conditional value function under the behavior policy: $m^{\circ, \pi}(S_t) = V_t^{\pi_t^b, \underline{\pi}_{t+1}} = \mathbb{E}_{\pi_t^b}[R_t + \gamma Q_{t+1}^\pi(S_{t+1}, A_{t+1}) \mid S_t]$. Under sequential unconfoundedness,

$$R_t + \gamma Q_{t+1}^\pi(S_{t+1}, A_{t+1}) - \epsilon(A_t) = Q_t^\pi(S_t, 0) + A_t \tau_t^\pi(S_t) \tag{3}$$

$$m_t^\pi(S_t) = \mathbb{E}_{\pi^b}[R_t + \gamma Q_{t+1}^\pi(S_{t+1}, A_{t+1}) \mid S_t] = Q_t^\pi(S_t, 0) + \pi^b(1 \mid S_t) \tau_t^\pi(S_t)$$

Hence,

$$R_t + \gamma Q_{t+1}^\pi(S_{t+1}, A_{t+1}) - m_t^\pi(S_t) = (A - \pi^b(1 \mid S_t)) \tau_t^\pi(S_t) + \epsilon_t. \tag{4}$$

---

**Algorithm 1** Dynamic Residualized Difference-of-Q-Evaluation

---

1: Given: $\pi^e$, evaluation policy; and for sample splitting, partition of $\mathcal{D}$ into $K$ folds, $\{\mathcal{D}_k\}_{k=1}^K$.

2: On $\mathcal{D}_k$, estimate $\hat{Q}^{\pi^e,k}(s,a)$ and behavior policy $\hat{\pi}_t^{b,k}(s)$. Evaluate the value function via integrating/summing $\hat{Q}$ over the empirical distribution of actions, $a \sim \pi^b$, so that

$$\hat{m}(s) = \mathbb{E}_{\pi_t^b}[R_t + \gamma \hat{Q}_{t+1}^{\pi^e}(S_{t+1}, A_{t+1}) \mid S_t = s].$$

3: **for** timestep $t = T, \ldots, 1$ **do**

4: $\quad \hat{\tau}_t \in \arg\min_\tau \left\{ \sum_{k=1}^K \sum_{i \in \mathcal{D}_k} (R_t^i + \gamma \hat{Q}_{t+1}^{\hat{\pi},-k}(S_{t+1}^i, A_{t+1}^i) - \hat{m}_t^{\hat{\pi},-k}(S_t^i) - \{A_t^i - \hat{\pi}_t^{b,-k}(S_t^i)\}\tau_t(S_t^i))^2 \right\}$

5: **end for**

---

**The loss function.** This motivates the approach based on (penalized) empirical risk minimization:

$$\tau_t(\cdot) \in \arg\min_\tau \left\{ \mathbb{E}\left[ \left( \{R_t + \gamma Q_{t+1}^\pi(S_{t+1}, A_{t+1}) - m_t^\pi(S_t)\} - \{A - \pi_t^b(1 \mid S_t)\} \cdot \tau(S_t) \right)^2 \right] \right\} \tag{5}$$

Again, so far we have discussed identification assuming the true $Q, m, \pi^b$ functions, etc. Next we discuss feasible estimation, and outside of this section we refer to the population-level true nuisance functions as $Q^{\pi,\circ}, m^{\pi,\circ}, \pi^{b,\circ}, \tau^{\pi,\circ}$.

**Extension to multiple actions.** So far we presented the method with $\mathcal{A} \in \{0, 1\}$ for simplicity, but all methods in this paper will extend to the multi-action case. We give the approach discussed in (Nie & Wager, 2021). For multiple actions, fix a choice $a_0 \in \mathcal{A}$, and for $a \in \mathcal{A} \setminus a_0$, define $\tau_t^\pi(s, a) \coloneqq \tau_{a,t}^\pi(s) = Q_t^\pi(s, a) - Q_t^\pi(s, a_0)$. For $k \in |\mathcal{A}|$, let $\pi^b(k \mid S_t) = P(A_t = k \mid S_t)$. Then a multivariate version of the argument gives the analogous loss function, where $\langle \cdot, \cdot \rangle$ is the dot product so that $\langle \vec{A} - \vec{\pi}^b(S_t), \tau_t^\pi(S_t) \rangle = \sum_{a \in \mathcal{A} \setminus a_0} \{(\mathbb{I}[A_t = a] - \pi^b(a \mid S_t))\tau_{a,t}^\pi(S_t)\}$:

$$\vec{\tau}_t(\cdot) \in \arg\min_\tau \{\mathbb{E}[(\{R_t + \gamma Q_{t+1}^\pi(S_{t+1}, A_{t+1}) - m_t^\pi(S_t)\} - \langle \vec{A} - \vec{\pi}^b(S_t), \vec{\tau}_t^\pi(S_t) \rangle)^2]\} \tag{6}$$

**Feasible estimation.** In practice, the $Q$ and behavior policy $\pi_b$ functions need to be estimated. Since they require auxiliary estimation but are not the final targets of analysis, they are termed "nuisance functions" in the causal ML literature. We introduce some notation before defining the full estimation algorithm. The nuisance vector is $\eta = [\{Q_t^\pi\}_{t=1}^T, \{m_t^\pi\}_{t=1}^T, \{\pi_t^b\}_{t=1}^T]$. We optimize a feasible version of the sequential loss minimization approach implied by eq. (6) with estimated nuisances (see Algorithm 1). Given an evaluation policy $\pi^e$, first fit estimates of the $Q$ function and the behavior policy. Then, evaluate the loss function in eq. (6) and estimate $\tau_t$.

The $Q$ function nuisance can be estimated with by-now standard approaches such as fitted-Q-evaluation (Le et al., 2019; Chakraborty & Murphy, 2014; Duan et al., 2021), minimum-distance estimation for conditional moment restrictions/GMM (Kallus & Uehara, 2019a), or the finite-horizon analogous version of a DR-learner. Estimating the behavior policy is a classic probabilistic classification or multi-class classification problem, though often the behavior policy is known by design.

**Cross-fitting.** We also introduce cross-fitting, which will differ slightly between policy evaluation and optimization: splitting the dataset $\mathcal{D}$ into $K$ many folds (preserving trajectories, i.e. randomizing over trajectory index $i$), and learning the nuisance function $\eta^{-k}$ on $\{\mathcal{D}_{k'}\}_{k' \in [K] \setminus k}$. (In scenarios with possible confusion we denote the nuisance function $\eta^{(-k)}$ instead.) In evaluating the loss-function, we evaluate the nuisance function $\eta^{-k}$ using data from the held-out $k$th fold. Given the cross-fitting procedure, we introduce the empirical squared loss function:

$$\hat{\mathcal{L}}_t(\tau, \eta) = \sum_{k=1}^K \sum_{i \in \mathcal{D}_k} (R_t^i + \gamma \hat{Q}_{t+1}^{\pi,-k}(S_{t+1}^i, A_{t+1}^i) - \hat{m}_t^{\pi,-k}(S_t^i) - \{A_t^i - \hat{\pi}_t^{b,-k}(1 \mid S_t^i)\}\tau_t(S_t^i))^2$$

and let the population loss function $\mathcal{L}_t(\tau, \eta)$ be the population expectation of the above.

**Policy optimization.** The sequential loss minimization approach also admits an policy optimization procedure. The policy is greedy with respect to the estimated $\tau_t$. We describe the algorithm in Algorithm 2. We use a slightly different cross-fitting approach for policy optimization. We introduce an additional fold, upon which we alternate estimation

---

**Algorithm 2** Dynamic Residualized Difference-of-Q Optimization

---

1: Given: Partition of $\mathcal{D}$ into 3 folds, $\{\mathcal{D}_k\}_{k=1}^3$.
2: Estimate $\hat{\pi}_t^b$ on $\mathcal{D}_1$.
3: At time T: Set $\hat{Q}_T(s,a) = 0$. Estimate $m_T = \mathbb{E}_{\pi^b}[R_T \mid S_T]$ on $\mathcal{D}_1$ and $\hat{\tau}_T$ on $\mathcal{D}_{k(T)}$, where $k(t) = 2$ if $t$ is odd and $k(t) = 3$ if $t$ is even.
   Optimize, for two actions $\hat{\pi}_T(s) = \mathbb{I}[\hat{\tau}_T(s) > 0]$.
4: **for** timestep $t = T-1, \ldots, 1$ **do**
5:   Estimate $Q^{\hat{\underline{\pi}}_{t+1}}$ on $\mathcal{D}_1$. Evaluate $m_t^{\hat{\underline{\pi}}_{t+1}}$. Estimate $\hat{\tau}_t^{\hat{\underline{\pi}}_{t+1}}$ on $\mathcal{D}_{k(t)}$ by minimizing the empirical loss:

$$\hat{\tau}_t(\cdot) \in \arg\min_\tau \sum_{i \in \mathcal{D}_{k(t)}} \left( R_t^i + \gamma \hat{Q}_{t+1}^{\hat{\pi},(1)}(S_{t+1}^i, A_{t+1}^i) - \hat{m}_t^{\hat{\pi},(1)}(S_t^i) - \{A_t^i - \hat{\pi}_t^{b,(1)}(S_t^i)\}\tau_t(S_t^i) \right)^2$$

6:   Optimal policy is greedy with respect to the difference-of-$Q$ function. For two actions, $\hat{\pi}_t(s) = \mathbb{I}[\hat{\tau}_t^{\hat{\underline{\pi}}_{t+1}}(s) > 0]$
   . Else for multiple actions, $\hat{\pi}_t(s) \in \arg\max_{a' \in \mathcal{A}\setminus a_0} \hat{\tau}_t^{\hat{\underline{\pi}}_{t+1}}(s, a')$ if $\max_{a' \in \mathcal{A}\setminus a_0} \hat{\tau}_t^{\hat{\underline{\pi}}_{t+1}}(s, a') > 0$, else $a_0$.
7: **end for**

---

of $\hat{\tau}_t$. So, overall we use three folds: one for estimating nuisance functions $\eta$, and the other two for estimating $\hat{\tau}_t^{\hat{\underline{\pi}}_{t+1}}$. On these two other folds, between every timestep, we alternate estimation of $\hat{\tau}_t$ on one of them, in order to break dependence between the estimated optimal forwards policy $\hat{\underline{\pi}}_{t+1}$ and $\hat{\tau}_t$ (and therefore the greedy policy $\hat{\pi}_t$).

Finally, note that the expectation of the empirical squared loss will not in general be an unbiased estimate of the true squared error, due to the squaring and the expectation over the next transition.

**Lemma 1** (Proxy Sequential R-learner loss: Excess Variance ).

$$\mathbb{E}[\hat{\mathcal{L}}_t(\tau^\pi, \eta)] - \mathcal{L}_t(\tau^\pi, \eta) = \text{Var}[\max_{a'} Q^\pi(S_{t+1}, a') \mid \pi_t^b]$$

Combining Lemma 1 with the identifying equations eq. (3), observe that the R-learner loss function is a "proxy loss" for estimation of the difference-of-Q function, with an additional variance term due to the squaring over next-transition stochasticity: $\mathbb{E}[\hat{\mathcal{L}}_t(\tau^\pi, \eta)] = \mathbb{E}[\text{Var}_{\pi_b}[A \mid S_t](\tau^{\circ,\pi}(S_t) - \tau_t^\pi(S_t))^2] + \text{Var}[\max_{a'} Q^\pi(S_{t+1}, a') \mid \pi_t^b]$. In particular, it estimates a *action-variance-weighted* version of the MSE for the difference-of-Q-function, upweighting states with a higher probability of seeing either action.

## 3 ANALYSIS

We study the improved statistical rates of convergence from orthogonal estimation for policy evaluation (Theorems 1 and 2) and show that this implies convergent policy optimization (Theorem 3). While Theorems 1 and 2 are applications of orthogonal statistical learning to our new estimand, we establish Neyman-orthogonality of our loss function (eq. (6)). Policy optimization is more challenging; the novelty of Theorem 3 is showing that estimation error from *policy-dependent nuisance functions* is of higher-order than the evaluation rates. Our analysis generally proceeds under the following assumptions.

**Assumption 2** (Independent and identically distributed trajectories). *We assume that the data was collected under a stationary behavior policy, i.e. not adaptively collected from a policy learning over time.*

**Assumption 3** (Boundedness). $V_t \leq B_V, \tau \leq B_\tau$

**Assumption 4** (Bounded transition density). *Transitions have bounded density: $P(s' \mid s, a) \leq c$. Let $d_\pi(s)$ denote the marginal state distribution under policy $\pi$. Assume that $d_{\pi_t^b}(s) < c$, for $t = 1, \ldots, T$.*

Assumption 2 highlights that we do not handle the challenges of dependent data or adaptive estimation, we can extend with standard tools. Assumption 4 permits translating error bounds learned under one state distribution to another, and it implies an all-policy concentrability coefficient. It is a common assumption and includes a wide class of MDPs (Munos & Szepesvári, 2008). Recent works improve upon all-policy concentrability though often with some pessimism or algorithmic modification. We leave improving upon this for future work.

Next we establish convergence rates for $\hat{\tau}^\pi$, depending on convergence rates of the nuisance functions. Broadly we follow the analysis of (Foster & Syrgkanis, 2019; Lewis & Syrgkanis, 2021) for orthogonal statistical learning. We

establish Neyman-orthogonality of the loss function in Appendix A.2, though this is overall fairly similar to prior work. The analysis considers some generic candidate $\hat{\tau}$ with small excess risk relative to the projection onto the function class, i.e. as might arise from an optimization algorithm with some approximation error. For a fixed evaluation policy $\pi^e$, define the projection of the true advantage function onto $\Psi^n$, $\tau_t^{\pi^e,n} = \arg\inf_{\tau_t \in \Psi_t^n} \|\tau_t - \tau_t^{\circ,\pi^e}\|_2$, and the error $\nu_t^{\pi^e} = \hat{\tau}_t^{\pi^e} - \tau_t^{n,\pi^e}$ of some estimate $\hat{\tau}_t^{\pi^e}$ to projection onto the function class:

**Theorem 1** (Policy evaluation ). *Suppose* $\{\sup_{s,t} \mathbb{E}[(A_t - \pi_t^b)(A_t - \pi_t^b) \mid s]\} \leq C$ *and Assumptions 1 to 4. Consider a fixed evaluation policy $\pi^e$. Consider any estimation algorithm that produces an estimate $\hat{\tau}^{\pi^e} = (\tau_1^{\pi^e}, \ldots, \tau_T^{\pi^e})$, with small plug-in excess risk at every t, with respect to any generic candidate $\tilde{\tau}^{\pi^e}$, at some nuisance estimate $\hat{\eta}$, i.e.,*

$$\mathcal{L}_{D,t}(\hat{\tau}_t^{\pi^e};\hat{\eta}) - \mathcal{L}_{D,t}(\tilde{\tau}_t^{\pi^e};\hat{\eta}) \leq \epsilon(\tau_t^n, \hat{\eta}).$$

*Let $\rho_t$ denote product error terms:*

$$\rho_t^{\pi^e}(\hat{\eta}) = B_\tau^2 \|(\hat{\pi}_t^b - \pi_t^{b,\circ})^2\|_u + B_\tau \|(\hat{\pi}_t^b - \pi_t^{b,\circ})(\hat{m}_t^{\pi^e} - m_t^{\pi^e,\circ})\|_u$$
$$+ \gamma(B_\tau \|(\hat{\pi}_t^b - \pi_t^{b,\circ})(\hat{Q}_{t+1}^{\pi^e} - Q_{t+1}^{\pi^e,\circ})\|_u + \|(\hat{m}_t^{\pi^e} - m_t^{\pi^e,\circ})(\hat{Q}_{t+1}^{\pi^e} - Q_{t+1}^{\pi^e,\circ})\|_u). \tag{7}$$

*Then, for $\sigma > 0$, and $u^{-1} + \overline{u}^{-1} = 1$,*

$$\frac{\lambda}{2}\|\nu_t^{\pi^e}\|_2^2 - \frac{\sigma}{4}\|\nu_t^{\pi^e}\|_{\overline{u}}^2 \leq \epsilon(\hat{\tau}_t^{\pi^e}, \hat{\eta}) + \frac{2}{\sigma}\left(\|(\tau^{\pi^e,\circ} - \tau_t^{\pi^e,n})\|_u^2 + \rho_t^{\pi^e}(\hat{\eta})^2\right).$$

In the above theorem, $\epsilon(\hat{\tau}_t^{\pi^e}, \hat{\eta})$ is the excess risk of the empirically optimal solution. Note that in our setting, this excess risk will be an approximation error incurred from the proxy loss issue described in Lemma 1.

The bias term is $\|(\tau^{\pi^e,\circ} - \tau_t^{\pi^e,n})\|_u^2$, which describes the model misspecification bias of the function class parametrizing $Q-$function contrasts, $\Psi$.

The product error terms $\rho_t^{\pi^e}(\hat{\eta})$ highlight the reduced dependence on individual nuisance error rates. We will instantiate the previous generic theorem for the projection onto $\Psi^n$, $\tau_t^{\pi^e,n}$, also accounting for the sample splitting. We will state the results with *local Rademacher complexity*, which we now introduce. For generic 1-bounded functions $f$ in a function space $f \in \mathcal{F}, f \in [-1,1]$, the local Rademacher complexity is defined as follows:

$$\mathscr{R}_n(\mathcal{F};\delta) = \mathbb{E}_{\epsilon_{1:n},X_{1:n}}\left[\sup_{f \in \mathcal{F}:\|f\|_2 \leq \delta} \frac{1}{n}\sum_{i=1}^n \epsilon_i f(X_i)\right]$$

The critical radius $\delta^2$ more tightly quantifies the statistical complexity of a function class, and is any solution to the so-called *basic inequality*, $\mathscr{R}_n(\mathcal{F};\delta) \leq \delta^2$. The star hull of a generic function class $\mathcal{F}$ is defined as $\text{star}(\mathcal{F}) = \{cf : f \in \mathcal{F}, c \in [0,1]\}$. Bounds on the critical radius of common function classes like linear and polynomial models, deep neural networks, etc. can be found in standard references on statistical learning theory, e.g. (Wainwright, 2019). We can obtain mean-squared error rates for policy evaluation via specializing Theorem 1 to the 2-norm and leveraging results from (Foster & Syrgkanis, 2019).

**Assumption 5** (Product error rates on nuisance function evaluation). *Fix an evaluation policy $\pi^e$. Suppose each of* $\mathbb{E}[\|(\hat{\pi}_t^b - \pi_t^{b,\circ})\|_2^2]$, $\mathbb{E}[\|(\hat{\pi}_t^b - \pi_t^{b,\circ})(\hat{m}_t^{\pi^e} - m_t^{\pi^e,\circ})\|_2^2]$, $\mathbb{E}[\|(\hat{\pi}_t^b - \pi_t^{b,\circ})(\hat{Q}_{t+1}^{\pi^e} - Q_{t+1}^{\pi^e,\circ})\|_2^2]$, *and* $\mathbb{E}[\|(\hat{m}_t^{\pi^e} - m_t^{\pi^e,\circ})(\hat{Q}_{t+1}^{\pi^e} - Q_{t+1}^{\pi^e,\circ})\|_2^2]$ *are of order* $O(\delta_{n/2}^2 + \|\tau_t^{\pi^e,\circ} - \tau_t^{\pi^e,n}\|_2^2)$.

*Denote the leading order of the product-error rate as $\rho$, i.e.* $O(\delta_{n/2}^2 + \|\tau_t^{\pi^e,\circ} - \tau_t^{\pi^e,n}\|_2^2) = O(n^{-\rho})$.

Assumption 5 summarizes both the product-error estimation rates and the misspecification error for $\tau$, $\|\tau_t^{\pi^e,\circ} - \tau_t^{\pi^e,n}\|_2^2$, in the rate term $\rho$. We further assume well-specification, leaving a more refined analysis for future work.

**Assumption 6** (Well-specified estimation of $\tau$). $\|\tau_t^{\pi^e,\circ} - \tau_t^{\pi^e,n}\|_2^2 = 0$.

Note that Assumption 6 pertains to the well-specification of the function class for estimating $\tau$, not $Q$. Meanwhile, Assumption 5 requires consistent estimation of the $Q$ function. Therefore we inherit potentially stringent structural restrictions for $Q$-function estimation such as Bellman completeness or linear Bellman completeness (Foster et al., 2021). Our orthogonal estimation enjoys the so-called "rate-double robustness" property, i.e. requiring only product-error $n^{-\frac{1}{2}}$ convergence of $Q, \pi_b$, but not the "mixed-bias" property of double-robustness wherein only one of $Q$ or $\pi_b$ need to be well-specified for unbiased estimation (Rotnitzky et al., 2021).

**Theorem 2** (MSE rates for policy evaluation)**.** *Suppose* $\left\{\sup_{s,t} \mathbb{E}[(A_t - \pi_t^b)(A_t - \pi_t^b) \mid s]\right\} \leq C$ *and Assumptions 1 to 4. Suppose Assumption 5 holds with rate $\rho$, i.e. Consider a fixed policy $\pi^e$. Then*

$$\mathbb{E}[\|\hat{\tau}_t^{\pi^e} - \tau_t^{\pi^e,\circ}\|_2^2] = O\left(\delta_{n/2}^2 + \|\tau_t^{\pi^e,\circ} - \tau_t^{\pi^e,n}\|_2^2\right)$$

For example, this states that estimation of $\pi_b, Q^{\pi^e}$ needs to be only $n^{-\frac{1}{4}}$ convergent to guarantee that the product error rate of Assumption 5 holds with rate $n^{-\frac{1}{2}}$. Standard methods for $Q$-function estimation alone would require $n^{-\frac{1}{2}}$ convergence directly. Working with the orthogonalized estimate results in the weaker product-error rate requirements included above. However, our estimating moments do include the $Q$ function nuisances, and quarter-root rates are required for estimating both the $Q$ and $\pi^b$ functions.

**Policy optimization.** Convergence of $\tau_t$ implies convergence in policy value. We quantify this with the *margin* assumption, which is a low-noise condition that quantifies the gap between regions of different optimal action (Tsybakov, 2004). It is commonly invoked to relate estimation error of plug-in quantities to decision regions, in this case the difference-of-Q functions to convergence of optimal decision values.

**Assumption 7** (Margin (Tsybakov, 2004))**.** *Assume there exist some constants $\alpha, \delta_0 > 0$ such that*

$$P\left(\max_a Q_t^*(s,a) - \max_{a' \in \mathcal{A} \setminus \arg\max_a Q_t^*(s,a)} Q_t^*(s,a') \leq \epsilon\right) = O\left(\varepsilon^\alpha\right), \forall t \in 1, \ldots, T$$

The probability density in Assumption 7 is respect to Lebesgue measure over the state space. Note that (Hu et al., 2024) establishes margin constants for linear and tabular MDPs; $\alpha = \infty$ for tabular MDPs, $\alpha = 1$ for linear MDPs where $Q^*$ is linear, and for nonlinear $Q^*$ under structural assumptions. The margin assumption allows us to translate advantage-function convergence rates to the policy value convergence rates, summarized in the following Lemma.

**Lemma 2** ( Advantage estimation error to policy value via margin.)**.** *Suppose Assumptions 1, 4 and 7 (margin assumption holds with $\alpha$). Suppose that with high probability $\geq 1 - n^{-\kappa}$ for any finite $\kappa > 0$, the following sup-norm convergence holds with some rate $b_* > 0$,*

$$\sup_{s \in \mathcal{S}, a \in \mathcal{A}} |\hat{\tau}_t^{\hat{\pi}_{t+1}}(s) - \tau_t^{\pi_{t+1}^*,\circ}(s)| \leq Kn^{-b*},$$

*then* $\left|\mathbb{E}[V_t^*(S_t) - V_t^{\hat{\pi}_{\hat{\tau}}}(S_t)]\right| \leq \frac{(1-\gamma^{T-t})}{1-\gamma} cK^2 n^{-b*(1+\alpha)} + O(n^{-\kappa}),$

*and* $\left\{\int \left(Q_t^*(s,\pi^*(s)) - Q_t^*(s,\hat{\pi}_{\hat{\tau}})\right)^2 ds\right\}^{1/2} \leq \frac{(1-\gamma^{T-t})}{1-\gamma} cK^2 n^{-b*(1+\alpha)} + O(n^{-\kappa}).$

*Else, assume that* $\|\hat{\tau}_t^n(s) - \tau_t^\circ(s)\|_2 \leq K\left(n^{-b_*}\right),$ *for some rate $b_* > 0$. Then*

$$\left|\mathbb{E}[V_t^*(S_t) - V_t^{\hat{\pi}_{\hat{\tau}}}(S_t)]\right| = O\left(n^{-b_*\left(\frac{2+2\alpha}{2+\alpha}\right)}\right), \text{ and } \left\{\int \left(Q_t^*(s,\pi^*(s)) - Q_t^*(s,\hat{\pi}_{\hat{\tau}})\right)^2 ds\right\}^{1/2} = O\left(n^{-b_*\left(\frac{2+2\alpha}{2+\alpha}\right)}\right).$$

Next we study policy optimization. Note that Lemma 2 relies on convergence of estimated difference-of-Q under the estimated-optimal policy to the true difference-of-Q under the optimal policy, while our earlier guarantees in Theorems 1 and 2 only imply convergence for the estimated-optimal policy alone. The remaining difficulty is the gap introduced by nuisance functions at estimated vs. true optimal policies; we show with induction and Assumption 7 that it is a *higher-order* estimation error. We state a simplified version of the full theorem in the main text; see the appendix for a more general statement.

**Theorem 3** (Policy optimization bound)**.** *Suppose Assumptions 1 to 4. Further, suppose that $Q^\circ$ satisfies Assumption 7 (margin) with $\alpha > 0$. Suppose the product error rate conditions of Assumption 5 hold for each $t$ for for $\hat{\pi}_{t+1}$, the data-optimal policies evaluated along the algorithm steps, with rates $\rho_t$ for each timestep. Suppose that then for $\hat{\pi}_t$, Theorem 2 holds. Denote the slowest such product-error-rate over timesteps as $\overline{\rho}_{1:T} = \min_t \{\rho_t\}$. Then,*

$$\|\hat{\tau}_t^{\hat{\pi}_{t+1}} - \tau_t^{\circ,\pi_{t+1}^*}\|_2 \leq O(n^{-\overline{\rho}_{1:T}}), \text{ and } \left|\mathbb{E}[V_1^{\pi^*}(S_1) - V_1^{\hat{\pi}_{\hat{\tau}}}(S_1)]\right| = O(n^{-\{\overline{\rho}_{1:T}\}\frac{2+2\alpha}{2+\alpha}}). \tag{8}$$

The main takeaway is that convergent policy evaluation admits convergent policy optimization, even though the nuisance functions evaluate an estimated-optimal, rather than population-optimal policy. This is because our method

**Algorithm 3** Stationary Infinite-Horizon Dynamic Residualized Difference-of-Q Optimization

1: Given: Partition of $\mathcal{D}$ into 3 folds, $\{\mathcal{D}_k\}_{k=1}^3$.

2: Estimate $\hat{\pi}_t^b$ on $\mathcal{D}_1$.

3: Estimate $\hat{Q}^{\hat{\pi}'}$ on $\mathcal{D}_1$ with offline policy optimization. Evaluate $m_t^{\hat{\pi}'}$.

4: Estimate $\hat{\tau}_t^{\hat{\pi}'}$ on $\mathcal{D}_{k(t)}$ by minimizing the empirical loss:

$$\hat{\tau}_t(\cdot) \in \arg\min_\tau \sum_{i \in \mathcal{D}_{k(t)}} \left( R_t^i + \gamma \hat{Q}^{\hat{\pi}'}(S_t^i, A_t^i) - \hat{m}_t^{\hat{\pi}'}(S_t^i) - \{A_t^i - \hat{\pi}_t^{b,(1)}(S_t^i)\}\tau_t(S_t^i) \right)^2$$

5: Policy optimization: For two actions, $\hat{\pi}_t(s) = \mathbb{I}[\hat{\tau}_t^{\hat{\pi}'}(s) > 0]$ . Else for multiple actions, $\hat{\pi}_t(s) \in \arg\max_{a' \in \mathcal{A} \setminus a_0} \hat{\tau}_t^{\hat{\pi}'}(s, a')$ if $\max_{a' \in \mathcal{A} \setminus a_0} \hat{\tau}_t^{\hat{\pi}'}(s, a') > 0$, else $a_0$.

Table 1: Validation: 1d example, linear FQE vs orthogonal $\tau$ estimation. Entries: Mean-squared error of estimated $\tau$ function, mean + 1.96 standard errors.

| Method (n) | $5.0 \cdot 10^1$ | $1.3 \cdot 10^3$ | $2.5 \cdot 10^3$ | $3.8 \cdot 10^3$ | $5.0 \cdot 10^3$ |
|---|---|---|---|---|---|
| FQE | $4 \cdot 10^{-2} \pm 2 \cdot 10^{-2}$ | $2 \cdot 10^{-3} \pm 10^{-3}$ | $10^{-3} \pm 10^{-4}$ | $10^{-3} \pm 10^{-4}$ | $4 \cdot 10^{-4} \pm 10^{-4}$ |
| OrthDiff-Q | $6 \cdot 10^{-2} \pm 3 \cdot 10^{-2}$ | $3 \cdot 10^{-3} \pm 10^{-3}$ | $2 \cdot 10^{-3} \pm 10^{-3}$ | $2 \cdot 10^{-3} \pm 10^{-3}$ | $2 \cdot 10^{-3} \pm 10^{-4}$ |

introduces auxiliary estimation at every timestep. Although this prevents the accumulation of faster exponentiated convergence, it also protects the method from error accumulation from policy-dependent nuisance estimation.

Our Theorem 3 presents the main technical novelty in the analysis, whereas Theorems 1 and 2 follow similar approaches as in previous literature. Because we estimate the optimal policy from time $t + 1$ forwards, this introduces challenges in showing benefits of orthogonality due to the policy-dependent nuisance, $Q_t^{\overline{\pi}_{t+1}}$ function. We show that the margin condition allows us to establish that the error due to policy-dependent nuisances, under an estimated optimal policy, is higher-order relative to the time-$t$ estimation of the advantage functions.

**Extension to stationary discounted infinite-horizon setting.** The identification argument extends to the stationary discounted infinite-horizon setting. For policy optimization, we make a small modification: instead of iterative optimization and estimation, we first conduct offline policy optimization to estimate the optimal policy and its $Q$ function, $\hat{\pi}'$ and $\hat{Q}^{\hat{\pi}'}$. We describe the algorithm in Algorithm 3. This can be done with a variety of methods that are common and popular in practice, such as DQN (Mnih et al., 2015), fitted-Q-iteration (Chen & Jiang, 2019), or other algorithms.

## 4 EXPERIMENTS

**1d validation.** In a very small 1d toy example (Sec 5.1, (Kallus & Uehara, 2019a)) we validate our method. See Appendix C of the appendix for more details.

**Adapting to structure in $\tau(s)$.** Recent research highlights the joint implications of blockwise conditional independence properties in RL, where some components are "exogenous" or irrelevant to rewards and actions (Wang et al., 2022; Wang et al.; Dietterich et al., 2018). Often these methods are designed around specific underlying assumptions, and therefore may be brittle under different substructures. Pretesting for the presence or absence of graphical restrictions warrants caution due to poor statistical properties. We advocate a different approach: by estimating the *difference-of-Q* functions, we can exploit statistical implications of underlying structure via sparse $\tau$, without vulnerability to assumptions on the underlying d.g.p.

We investigate the benefits of targeting estimation of the difference-of-Qs in two different graphical substructures, replicated in Section 1, proposed in Zhou (2024b); Dietterich et al. (2018). Orthogonal causal contrast estimation is robust under noisy nuisance functions, as confirmed by our theory, and that it can adapt to a variety of structures.

First we describe the modified Reward-Filtered DGP (left, fig. 2) of (Zhou, 2024b). In the DGP, $|\mathcal{S}| = 100$ though the first $|\rho| = 15$ dimensions are the reward-relevant sparse component, where $\rho$ is the indicator vector of the sparse support, and $\mathcal{A} = \{0, 1\}$. The reward and states evolve according to $r_t(s, a) = \beta^\top \phi_t(s, a) + a * \sum_{k=1}^5 s_k/2 + \epsilon_r$, $s_{t+1}(s, a) = M_a s + \epsilon_s$, satisfying the graphical restrictions of Section 1. Therefore the transition matrices satisfy

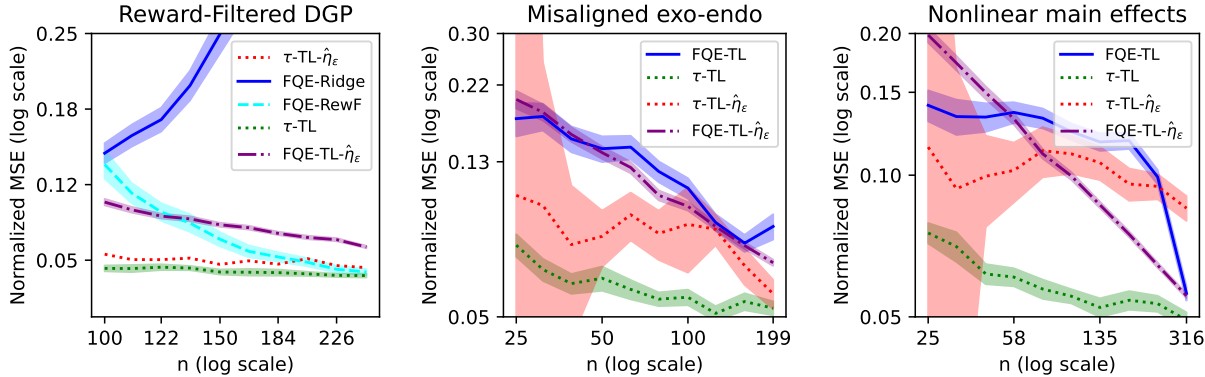

Figure 2: Adapting to structure. Interacted setting where $E[M_1 - M_0] = 0.1 \cdot I$.

the blockwise form $M_a = \begin{bmatrix} M_a^{\rho \to \rho} & 0 \\ M_a^{\rho \to \rho_c} & M_a^{\rho_c \to \rho_c} \end{bmatrix}$. We generate the coefficient matrices $M_0, M_1$ with independent normal random variables $\sim N(0.2, 1)$. The nonzero mean ensures the beta-min condition. We normalize $M_a^{\rho \to \rho}$ to have spectral radius 1, then introduce mild instability in the exogenous component by dividing $M_a^{\rho_c \to \rho_c}$ by $0.8$ times the largest eigenvalue. Therefore, recovering the sparse component is stable while including distracting dimensions destabilizes. The zero-mean noise terms are normally distributed with standard deviations $\sigma_s = 0.3, \sigma_r = 0.5$. In the estimation, we let $\phi(s, a) = \langle s, s\dot a, 1 \rangle$ is the interacted state-action space. The behavior policy is a mixture of logistic, with coefficients $\sim N(0, 0.3)$, and 20% probability of uniform random sampling. The evaluation policy is logistic, with coefficients $\sim \text{Unif}[-0.5, 0.5]$. (We fix the random seed).

In Figure 2 we compare against baselines. In blue is FQE-Ridge, i.e. naive fitted-Q-evaluation with ridge regression. In dotted cyan is FQE-RF, the reward-filtered method of (Zhou, 2024b). Next we have two variants of our framework: in dotted green $\tau$-TL which uses reward-based thresholding to estimate $\tau$ on the recovered support, and dotted-red $\tau$-TL-$\hat{\eta}_\epsilon$, the same method with sample splitting with noisy nuisances. With $\tau$-TL-$\hat{\eta}_\epsilon$, we investigate semi-synthetic settings with noisy nuisance functions by adding $N(0, n^{-1/4})$ noise to nuisance function predictions. For comparison to illustrate a setting with slow nuisance function convergence, we also include in dot-dashed purple FQE-TL-$\hat{\eta}_\epsilon$, which adds $n^{-1/4}$ noise to the oracle difference-of-$Q$ function (estimated with LASSO). For our methods, we solve the loss function minimization exactly with CVXPY (and l1 norm regularization).

We describe the results left to right. We display the mean over $100$ replications (fixing the coefficient matrices and vectors, etc. with the same random seed); except for sample-splitting where we display the median. (With small $n$, sample splitting suffers finite-sample issues of small data splits, though this vanishes as $n$ increases). The y-axis is the normalized MSE (we divide by the square of the range of the true difference of $Q$s), and the x axis is the number of episodes, on a log scale. First on the left, we consider the previously mentioned reward-filtered DGP. The tailored method of (Zhou, 2024a) is well-specified. For the reward-filtered DGP, we compare against FQE ridge regression, which we also use as a nuisance estimator for our approach. When compared to oracle-sparse difference-of-Q estimation, naive ridge FQE even diverges. However, our methods with thresholded LASSO do well, even if we plug-in the nuisance $Q$ function estimated with Ridge regression. We do see that orthogonality can provide an estimation rate speed up in the case of noisy nuisances, i.e. the red-dotted line with noisy nuisance functions indicates the robustness of orthogonal estimation to slower nuisance function convergence. (The additional sample splitting leads to transient small-data issues and affects the constants in the rate of convergence, but not the rate.)

Next we slightly modify the graphical structure. Our methods adapt to the underlying sparsity in the difference-of-Q functions, *even if* the exact graphical independences differ. In all the experiments, naive cross-validation does poorly. This is consistent with the research literature since cross-validation for predictive error doesn't ensure support recovery, unlike thresholded LASSO, and suffers additional challenges of hyperparameter tuning in offline RL.

In "Misaligned endo-exo", we follow the same data-generating process as for the "Reward-Filtered DGP" described earlier, but we change the blockwise conditional independences to follow the exogeneous-endogenous model of (Dieterich et al., 2018) (see Section 1, right). We additionally added dense rewards to the reward vector, adding

Table 2: Performance comparison on different sample numbers under the nonlinear setting.

| Method (n) | 100 | 200 | 400 | 600 | 800 |
|:---:|:---:|:---:|:---:|:---:|:---:|
| FQE | $2.367 \pm 2.157$ | $0.587 \pm 0.772$ | $1.157 \pm 2.219$ | $1.793 \pm 1.618$ | $4.123 \pm 3.901$ |
| DiffQ | $2.212 \pm 2.376$ | $0.415 \pm 0.463$ | $1.228 \pm 1.831$ | $1.929 \pm 2.126$ | $2.440 \pm 1.912$ |
| DiffQ+MI | $2.104 \pm 2.392$ | $0.280 \pm 0.222$ | $1.179 \pm 1.840$ | $1.286 \pm 1.123$ | $2.342 \pm 1.812$ |

$\beta_{dense}^{\top} \phi_t(s,a)$ where the entries of $\beta_{dense}$ are 1 w.p. 0.9. In this setting, reward sparsity of $R(s,a), a \in \{0,1\}$ alone does not recover the sparse component. Reward-filtered thresholded LASSO is simply misspecified and does very poorly (MSE greater than graph limits). Likewise, in small samples, vanilla thresholded LASSO FQE (FQE-TL, dark-blue) simply includes too many extraneous dimensions which destabilize estimation. In contrast for small-data regimes, imposing thresholded LASSO *on the difference of Q functions* remains stable.

The final DGP of "Nonlinear main effects" introduces nonlinear main effects: again we generate a 50% dense vector $\beta_{dense}$ and we add $s^{\top}\beta_{dense} + 3\sin(\pi s_{49}s_{48}) + 0.5(s_{49} - 0.5)^2 + 0.5(s_{48} - 0.5)^2$. (These nonlinear main effects are therefore disjoint from the sparse set and sparse difference-of-Q terms). For small $n$, FQE wrongly includes extraneous dimensions that destabilize estimation, and our methods directly estimating $\tau$ with reward-thresholded-LASSO outperform naive FQE with thresholded-LASSO regularization for small data sizes.

**Extending to nonlinear settings: mutual information regularization.** Our experiments generally indicate that regularization that achieves support recovery is necessary. To illustrate how the loss function approach permits more complex parametrizations, we now consider neural-nets and introduce a heuristic regularizer based on mutual information regularization. We use a cartpole-with-distractors environment, appending additional autoregressive noise variables to the state representation of CartPole (Brockman et al., 2016). We adapt a specific cartpole-with-distractors experimental setup found in Hao et al. (2024), which focuses on off-policy evaluation with abstractions, so the methods are not comparable.[1] Given this environment, which is a long finite-horizon environment with time-homogenous transitions, we learn this as a $\gamma = 0.99$ discounted infinite-horizon problem and pool the data across timesteps.

We explore the use of mutual information (MI) as a regularization term to optimize our loss function over a simpler representation recovers information related to the loss function, and discards irrelevant information that is unrelated to the proxy loss for the difference-of-Q functions. We use a mutual information regularizer (MIR), $\hat{\mathcal{L}}_{MI}(\phi, \theta)$, to encourage decomposing the state $S$ into independent nonlinear representations $X_c^{\phi}, X_a^{\theta}$, parametrized respectively by $\phi, \theta$. As mutual information quantifies the dependency between two variables, it equals zero if and only if the variables are (marginally) independent. We parametrize the difference-of-Q function as $\tau(X_c^{\phi})$, depending only on the the *confounding* information $X_c^{\phi}$ which is relevant to the difference-of-Q loss function, while the *auxiliary* information $X_a^{\theta}$ is independent of the loss function. We also add a reconstruction loss function $\hat{\mathcal{L}}_{rec}(\phi, \theta)$ which ensures that these two representations jointly recover the state. These additional loss functions are weighted by hyperparameters $\lambda_m, \lambda_r$.

$$\hat{\mathcal{L}}^{nl}(\tau, \eta; \phi, \theta) = \hat{\mathcal{L}}(\tau_{\phi}, \eta) + \lambda_m \hat{\mathcal{L}}_{MI}(\phi, \theta) + \lambda_r \hat{\mathcal{L}}_{rec}(\phi, \theta),$$
$$\text{where } \hat{\mathcal{L}}_{MI} = |\hat{I}(X_a^{\phi}; X_c^{\theta})|, \hat{\mathcal{L}}_{rec} = \mathbb{E}[(X_a^{\phi} + X_c^{\theta} - S)^2]$$

Estimating mutual information is challenging. We use a recently developed mutual information neural-networks based estimator, abbreviated MINE (Belghazi et al., 2018a). (See Appendix C for more details). MINE defines a neural information measure $I_{\Lambda}(X_a, X_c) = \sup_{\lambda \in \Lambda} \mathbb{E}_{\mathbb{P}_{X_a X_c}}[T_{\lambda}] - \log(\mathbb{E}_{\mathbb{P}_{X_a} \otimes \mathbb{P}_{X_c}}[e^{T_{\lambda}}])$. Usually MI requires functional form access to probability densities, though only samples from the joint distribution in machine learning-based methods are available. MINE empirically uses the samples of joint $\mathbb{P}_{X_a X_c}$ and marginals $\mathbb{P}_{X_a} \otimes \mathbb{P}_{X_c}$ distribution and obtains the empirical distribution $\hat{\mathbb{P}}^{(n)}$ associated to $n$ i.i.d. samples.

We illustrate how our method can improve upon naive FQE (learned with neural nets) learned on the full state space. We compare to an oracle difference-of-Q function obtained by differencing $Q$ estimates from FQE from a large dataset, $n = 2000$, trained only on the original 4-dim state space without distractors. We compare to our DiffQ estimation, parametrizing $\tau$ with neural nets, and a regularized version. Model selection in offline RL is somewhat of an open problem, we leave investigating approaches for future work.

---

[1] In the language of abstractions, the reward-filtered data-generating process is an example of sub-component structure where linear sparse estimation recovers a *model-irrelevant* abstraction.

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

# A PROOFS

## A.1 PRELIMINARIES

**Lemma 3** (Excess Variance ).

$$\mathbb{E}[\hat{\mathcal{L}}_t(\tau,\eta)] - \mathcal{L}_t(\tau,\eta) = \mathrm{Var}[\max_{a'} Q\left(S_{t+1}, a'\right) \mid \pi_t^b]$$

*Proof.*
$$\mathbb{E}[\hat{\mathcal{L}}_t(\tau,\eta)]$$

$$= \mathbb{E}[\left(\{R_t + \gamma Q_{t+1}^{\pi^e}(S_{t+1}, A_{t+1}) - m_t^{\pi}(S_t)\} \pm \mathbb{E}[\gamma Q_{t+1}^{\pi^e} \mid S_t, \pi_t^b] - \{A - \pi_t^b(1 \mid S_t)\} \cdot \tau(S_t)\right)^2]$$

$$= \mathbb{E}[\left(\{R_t + \gamma \mathbb{E}[Q_{t+1}^{\pi^e} \mid S_t, \pi_t^b] - m_t^{\pi}(S_t)\} - \{A - \pi_t^b(1 \mid S_t)\} \cdot \tau(S_t) + \gamma(Q_{t+1}^{\pi^e}(S_{t+1}, A_{t+1}) - \mathbb{E}[Q_{t+1}^{\pi^e} \mid S_t, \pi_t^b])\right)^2]$$

$$= \mathbb{E}[\left(\{R_t + \gamma \mathcal{T} Q_{t+1}^{\pi^e} - m_t^{\pi}\} - \{A - \pi_t^b(1 \mid S_t)\} \cdot \tau(S_t)\right)^2] \qquad \text{(squared loss of identifying moment)}$$

$$+ \mathbb{E}[\gamma(Q_{t+1}^{\pi^e}(S_{t+1}, A_{t+1}) - \mathbb{E}[Q_{t+1}^{\pi^e} \mid S_t, \pi_t^b])^2] \qquad \text{(residual variance of } Q_t(s,a) - R_t(s,a))$$

$$+ \mathbb{E}[\left\{R_t + \gamma \mathbb{E}[Q_{t+1}^{\pi^e} \mid S_t, \pi_t^b] - m_t^{\pi}(S_t) - \{A - \pi_t^b(1 \mid S_t)\} \cdot \tau(S_t)\right\} \cdot \gamma(Q_{t+1}^{\pi^e}(S_{t+1}, A_{t+1}) - \mathbb{E}[Q_{t+1}^{\pi^e} \mid S_t, \pi_t^b])]$$

Note the last term $= 0$ by iterated expectations and the pull-out property of conditional expectation. $\qquad\square$

## A.2 ORTHOGONALITY

Below we will omit the $\pi$ superscript; the analysis below holds for any valid $\pi$. Define $\nu_t = \hat{\tau}_t - \tau_t^n, \nu_t^{\circ} = \hat{\tau}_t - \tau_t^{\circ}$. We define for any functional $L(f)$ the Frechet derivative as:

$$D_f L(f)[\nu] = \left.\frac{\partial}{\partial t} L(f + t\nu)\right|_{t=0}$$

Higher order derivatives are denoted as $D_{g,f} L(f, g)[\mu, \nu]$.

**Lemma 4** (Universal Orthogonality).

$$D_{\eta,\tau_t} \mathcal{L}_t(\tau_t^n; \tau_{t+1}^n, \eta^*)[\eta - \eta^*, \nu_t] = 0$$

*Proof of Lemma 4.* For brevity, for a generic $f$, let $\{f\}_\epsilon$ denote $f + \epsilon(f - f^{\circ})$. Then the first Frechet derivatives are:

$$\frac{d}{d\epsilon_\tau} \mathcal{L}_t(\tilde{\tau}, \eta^{\circ})[\tau - \tilde{\tau}, \eta - \eta^{\circ}] = \mathbb{E}\left[\left\{R_t + \gamma\{Q_{t+1}^{\pi^e,\circ}\}_\epsilon - \{m_t^{\pi^e,\circ}\}_\epsilon - (A_t - \{\pi_t^{b\circ}\}_\epsilon)\tau\right\}(A_t - \{\pi_t^{b,\circ}\}_\epsilon)(\tilde{\tau} - \tau)\right]$$

$$\left.\frac{d}{d\epsilon_e}\frac{d}{d\epsilon_\tau} \mathcal{L}_t(\tilde{\tau}, \eta^{\circ})[\eta - \eta^{\circ}, \tau - \tau]\right|_{\epsilon=0}$$
$$= \mathbb{E}\left[\left(\pi_t^b - \pi_t^{b,\circ}\right)\tau(\tau - \tilde{\tau})(A_t - e_t)\right] + \mathbb{E}\left[\left\{R + \gamma Q_{t+1}^{\pi^e} - m_t^{\pi^e,\circ} - (A_t - e_t)\right\}(\tau - \tilde{\tau}) \cdot -(e_t - e_t^{\circ})\right]$$
$$= 0$$

$$\left.\frac{d}{d\epsilon_{Q_{t+1}}}\frac{d}{d\epsilon_\tau} \mathcal{L}_t(\tilde{\tau}, \eta^{\circ})[\eta - \eta^{\circ}, \tau - \tilde{\tau}]\right|_{\epsilon=0}$$
$$= \mathbb{E}[\gamma(Q_{t+1}^{\pi^e} - Q_{t+1}^{\pi^e,\circ})(A_t - \pi_t^{b,\circ})(\tau_t - \tilde{\tau}_t)]]$$
$$= 0$$

$$\left.\frac{d}{d\epsilon_{m_t}}\frac{d}{d\epsilon_\tau} \mathcal{L}_t(\tilde{\tau}, \eta^{\circ})[\eta - \eta^{\circ}, \tau - \tilde{\tau}]\right|_{\epsilon=0}$$
$$= \mathbb{E}[-(m_t^{\pi^e} - m_t^{\pi^e,\circ})(A_t - \pi_t^{b,\circ})(\tau_t - \tilde{\tau}_t)]$$
$$= 0$$

$\qquad\square$

**Lemma 5** (Second order derivatives). *For $Q_{t+1}, Q_{t+1}^\circ$ evaluated at some fixed policy $\pi^e$:*

$$D_{\eta_t,\eta_t}\mathcal{L}_t[\hat{\eta}_t - \eta_t^\circ, \hat{\eta}_t - \eta_t^\circ]$$

$$= \mathbb{E}\left[\tau_t^2\left(\hat{\pi}_t^b - \pi_t^{b,\circ}\right)^2\right] + \mathbb{E}\left[(\hat{\pi}_t^b - \pi_t^{b,\circ})\tau_t(\hat{m}_t - m_t^\circ)\right] + \mathbb{E}\left[(\hat{\pi}_t^b - \pi_t^{b,\circ})\tau_t\gamma(\hat{Q}_{t+1} - Q_{t+1}^\circ)\right]$$

$$- \mathbb{E}\left[(\hat{m}_t - m_t^\circ)\gamma\left(\hat{Q}_{t+1} - Q_{t+1}^\circ\right)\right]$$

*Proof of Lemma 5.* Below, the evaluation policy $\pi^e$ is fixed and omitted for brevity. Note that

$$D_e\mathcal{L}_D[\hat{e} - e^\circ] = \mathbb{E}[(R_t + \gamma Q_{t+1} - \pi^{b\top}Q_t + (A - \pi_t^b)\tau_t)(-\tau_t)(\hat{e} - e^\circ)]$$

$$D_{m_t}\mathcal{L}_D[\hat{m}_t - m_t^\circ] = \mathbb{E}[(R_t + \gamma Q_{t+1} - \pi^{b\top}Q_t + (A - \pi_t^b)\tau_t)(-1) * (m_t - m^\circ)]$$

By inspection, note that the nonzero terms of the second-order derivatives are as follows:

$$D_{\pi_t^b,\pi_t^b}\mathcal{L}_t[\hat{\pi}_t^b - \pi_t^{b,\circ}, \hat{\pi}_t^b - \pi_t^{b,\circ}] = \mathbb{E}\left[\tau_t^2\left(\hat{\pi}_t^b - \pi_t^{b,\circ}\right)^2\right]$$

$$D_{m_t,Q_{t+1}}\mathcal{L}_t[\hat{Q}_{t+1} - Q_{t+1}^\circ, \hat{m}_t - m_t^\circ] = \mathbb{E}\left[-(\hat{m}_t - m_t^\circ)\gamma\left(\hat{Q}_{t+1} - Q_{t+1}^\circ\right)\right]$$

$$D_{m_t,\pi_t^b}\mathcal{L}_t[\hat{\pi}_t^b - \pi_t^{b,\circ}, \hat{m}_t - m_t^\circ] = \mathbb{E}\left[(\hat{\pi}_t^b - \pi_t^{b,\circ})\tau_t(\hat{m}_t - m_t^\circ)\right]$$

$$D_{Q_{t+1},\pi_t^b}\mathcal{L}_t[\hat{\pi}_t^b - \pi_t^{b,\circ}, \hat{Q}_{t+1} - Q_{t+1}^\circ] = \mathbb{E}\left[(\hat{\pi}_t^b - \pi_t^{b,\circ})\tau_t\gamma(\hat{Q}_{t+1} - Q_{t+1}^\circ)\right]$$

By the chain rule for Frechet differentiation, we have that

$$D_{\eta_t,\eta_t}\mathcal{L}_t[\hat{\eta}_t - \eta_t^\circ, \hat{\eta}_t - \eta_t^\circ] = D_{\pi_t^b,\pi_t^b}\mathcal{L}_t[\hat{\pi}_t^b - \pi_t^{b,\circ}, \hat{\pi}_t^b - \pi_t^{b,\circ}]$$

$$+ D_{m_t,\pi_t^b}\mathcal{L}_t[\hat{\pi}_t^b - \pi_t^{b,\circ}, \hat{m}_t - m_t^\circ] + D_{Q_{t+1},\pi_t^b}\mathcal{L}_t[\hat{\pi}_t^b - \pi_t^{b,\circ}, \hat{Q}_{t+1} - Q_{t+1}^\circ] + D_{m_t,Q_{t+1}}\mathcal{L}_t[\hat{Q}_{t+1} - Q_{t+1}^\circ, \hat{m}_t - m_t^\circ]$$

$$\square$$

## A.3 PROOF OF SAMPLE COMPLEXITY BOUNDS

*Proof of Lemma 2.*

$$V_t^*(s) - V_t^{\pi_{\hat{\tau}}}(s) = V_t^*(s) - V_t^{\pi_{\hat{\tau}}}(s) \pm Q^{\pi^*}(s, \pi_{\hat{\tau}})$$

$$= Q_t^*(s, \pi^*(s)) - Q_t^*(s, \hat{\pi}_{\hat{\tau}}) + Q_t^*(s, \hat{\pi}_{\hat{\tau}}) - V_t^{\hat{\pi}_{\hat{\tau}}}(s)$$

$$\leq \gamma\mathbb{E}_{\hat{\pi}_t}\left[V_{t+1}^{\pi^*} - V_{t+1}^{\hat{\pi}_{\hat{\tau}}} \mid s\right] + Q_t^*(s, \pi^*(s)) - Q_t^*(s, \hat{\pi}_{\hat{\tau}})$$

Therefore for any $t$ and Markovian policy $\pi$ inducing a marginal state distribution:

$$\mathbb{E}[V_t^*(s)] - \mathbb{E}[V_t^{\pi_{\hat{\tau}}}(s)] \leq \gamma\mathbb{E}\left[\mathbb{E}_{\hat{\pi}_t}[V_{t+1}^{\pi^*} - V_{t+1}^{\hat{\pi}_{\hat{\tau}}} \mid s]\right] + \mathbb{E}[Q_t^*(s, \pi^*) - Q_t^*(s, \hat{\pi}_{\hat{\tau}})] \tag{9}$$

Assuming bounded rewards implies that $P(s_{t+1} \mid s, a) \leq c$, which remains true under the state-action distribution induced by any Markovian policy $\pi(s, a)$, including the optimal policy. Therefore the second term of the above satisfies:

$$\mathbb{E}_\pi[Q_t^*(s_t, \pi^*) - Q_t^*(s_t, \hat{\pi}_{\hat{\tau}})] \leq c\int\{Q_t^*(s, \pi^*) - Q_t^*(s, \hat{\pi}_{\hat{\tau}})\}\,ds, \tag{10}$$

and fixing $t = 1$, we obtain:

$$\mathbb{E}[Q_1^*(s_1, \pi^*) - Q_1^*(s_1, \hat{\pi}_{\hat{\tau}})] \leq c\int\{Q_1^*(s, \pi^*) - Q_1^*(s, \hat{\pi}_{\hat{\tau}})\}\,ds.$$

Next we continue for generic $t$ and bound the right hand side term of eq. (10).

First we suppose we have a high-probability bound on $\ell_\infty$ convergence of $\hat{\tau}$. Define the good event

$$\mathcal{E}_g = \left\{ \sup_{s \in \mathcal{S}, a \in \mathcal{A}} |\hat{\tau}^{\hat{\pi}_{t+1}}(s) - \tau^{\pi^*_{t+1}, \circ}_t(s)| \le K n^{-b*} \right\}$$

A maximal inequality gives that $P(\mathcal{E}_g) \ge 1 - n^{-\kappa}$. We have that

$$\int \{Q^*_t(s, \pi^*(s)) - Q^*_t(s, \hat{\pi}_{\hat{\tau}})\} \, ds = \int \{Q^*_t(s, \pi^*(s)) - Q^*_t(s, \hat{\pi}_{\hat{\tau}})\} \, \mathbb{I}\left[\mathcal{E}_g\right] ds + \int \{Q^*_t(s, \pi^*(s)) - Q^*_t(s, \hat{\pi}_{\hat{\tau}})\} \, \mathbb{I}\left[\mathcal{E}^c_g\right] ds$$

$$(11)$$

Assuming boundedness, the bad event occurs with vanishingly small probability $n^{-\kappa}$, which bounds the second term of eq. (15).

For the first term of eq. (15), note that on the good event, if mistakes occur such that $\pi^*_t(s) \neq \hat{\pi}_t(s)$, then the true contrast function is still bounded in magnitude by the good event ensuring closeness of the estimate, so that $\left|\tau^{\pi^*_{t+1}, \circ}_t(s)\right| \le 2K n^{-b*}$. And if no mistakes occur, at $s$ the contribution to the integral is 0. Denote the mistake region as

$$\mathcal{S}_m = \{s \in \mathcal{S} \colon \left|\tau^{\pi^*_{t+1}, \circ}_t(s)\right| \le 2K n^{-b*}\}$$

Therefore

$$\int \{Q^*_t(s, \pi^*(s)) - Q^*_t(s, \hat{\pi}_{\hat{\tau}})\} \, ds \le \int_{s \in \mathcal{S}_m} \{Q^*_t(s, \pi^*(s)) - Q^*_t(s, \hat{\pi}_{\hat{\tau}})\} \, \mathbb{I}\left[s \in \mathcal{S}_m\right] \mathbb{I}\left[\mathcal{E}_g\right] ds + O(n^{-\kappa}) \quad (12)$$

Note also that (for two actions), if *action* mistakes occur on the good event $\mathcal{E}_g$, the difference of $Q$ functions must be near the decision boundaries so that we have the following bound on the integrand:

$$|Q^*(s, \pi^*) - Q^*(s, \hat{\pi})| \le |\tau^{\pi^*_{t+1}, \circ}| \le 2K n^{-b*}. \quad (13)$$

Therefore,

$$\int \{Q^*_t(s, \pi^*(s)) - Q^*_t(s, \hat{\pi}_{\hat{\tau}})\} \, ds \le O(n^{-\kappa}) + K n^{-b*} \int \mathbb{I}\left[s \in \mathcal{S}_m\right] ds$$

$$\le O(n^{-\kappa}) + (K n^{-b*})(K n^{-b*\alpha})$$

$$= O(n^{-\kappa}) + (K^2 n^{-b*(1+\alpha)}) \quad (14)$$

where the first inequality follows from the above, and the second from assumption 7 (margin).

Combining eqs. (9) and (14), we obtain:

$$\mathbb{E}[V^*_t(S_t)] - \mathbb{E}[V^{\hat{\pi}_{\hat{\tau}}}_t(S_t)] \le \sum_{t=1}^{T} \gamma^t c \left\{ \int Q^{\hat{\pi}_{\hat{\tau}}}_t(s, \pi^*(s)) - Q^{\hat{\pi}_{\hat{\tau}}}_t(s, \hat{\pi}_{\hat{\tau}}) ds \right\}$$

$$\le \frac{(1 - \gamma^T)}{1 - \gamma} cT \{O(n^{-\kappa}) + (K^2 n^{-b*(1+\alpha)})\}$$

We also obtain analogous results for norm bounds:

$$\left\{ \int (Q^*_t(s, \pi^*(s)) - Q^*_t(s, \hat{\pi}_{\hat{\tau}}))^u \, ds \right\}^{1/u}$$

$$\le \left\{ \int_{s \in \mathcal{S}_m} (Q^*_t(s, \pi^*(s)) - Q^*_t(s, \hat{\pi}_{\hat{\tau}}))^u \mathbb{I}\left[s \in \mathcal{S}_m\right] \mathbb{I}\left[\mathcal{E}_g\right] ds \right\}^{1/u} + O(n^{-\kappa})$$

$$\le \frac{(1 - \gamma^T)}{1 - \gamma} cT \{O(n^{-\kappa}) + (K^2 n^{-b*(1+\alpha)})\}$$

The results under an integrated risk bound assumption on convergence of $\tau$ follow analogously as (Shi et al., 2022), which we also include for completeness. For a given $\varepsilon > 0$, redefine the mistake region parametrized by $\epsilon$:

$$\mathcal{S}_\epsilon = \left\{ \max_a Q^*(s, a) - Q^*(s, \hat{\pi}(s)) \le \varepsilon \right\}.$$

Again we obtain the bound by conditioning on the mistake region:

$$\int \{Q_t^*(s, \pi^*(s)) - Q_t^*(s, \hat\pi_{\hat\tau})\} \, ds = \int \{Q_t^*(s, \pi^*(s)) - Q_t^*(s, \hat\pi_{\hat\tau})\} \, \mathbb{I}\,[\mathcal{S}_\epsilon] \, ds + \int \{Q_t^*(s, \pi^*(s)) - Q_t^*(s, \hat\pi_{\hat\tau})\} \, \mathbb{I}\,[\mathcal{S}_\epsilon^c] \, ds \tag{15}$$

Using similar arguments as earlier, we can show by Assumption 7:

$$\int \{Q_t^*(s, \pi^*(s)) - Q_t^*(s, \hat\pi_{\hat\tau})\} \, \mathbb{I}(s \in \mathcal{S}_*) ds \leq \varepsilon \int_x \mathbb{I}(s \in \mathcal{S}_*) ds = O\left(\varepsilon^{1+\alpha}\right).$$

As previously argued, we can show mistakes $\pi_t^*(s) \neq \hat\pi_t(s)$ occur only when

$$\max_a Q_t^*(s, a) - Q^*(s, \hat\pi(s)) \leq 2 \left| \hat\tau^{\hat\pi_{t+1}}(s) - \tau^{\pi_{t+1}^*, \circ}(s) \right|. \tag{16}$$

It follows that

$$\int \{Q_t^*(s, \pi^*(s)) - Q_t^*(s, \hat\pi_{\hat\tau})\} \, \mathbb{I}\,[s \in \mathcal{S}_\epsilon^c] \, ds$$

$$\leq \mathbb{E} \int \frac{4 \left| \hat\tau^{\hat\pi_{t+1}}(s) - \tau^{\pi_{t+1}^*, \circ}(s) \right|^2}{\{Q_t^*(s, \pi^*(s)) - Q_t^*(s, \hat\pi_{\hat\tau})\}} \mathbb{I}\,[s \in \mathcal{S}_\epsilon^c] \, ds$$

$$\leq \frac{4}{\varepsilon} \int \left| \hat\tau^{\hat\pi_{t+1}}(s) - \tau^{\pi_{t+1}^*, \circ}(s) \right|^2 ds = O\left(\varepsilon^{-1} |\mathcal{I}|^{-2b_*}\right).$$

Combining this together with (E.106) and (E.107) yields that

$$\int \{Q_t^*(s, \pi^*(s)) - Q_t^*(s, \hat\pi_{\hat\tau})\} \, ds = O\left(\varepsilon^{1+\alpha}\right) + O\left(\varepsilon^{-1} |\mathcal{I}|^{-2b_*}\right).$$

The result follows by choosing $\varepsilon = n^{-2b_*/(2+\alpha)}$ to balance the two terms.

For the norm bound, the first term is analogously bounded as $O\left(\varepsilon^{1+\alpha}\right)$:

$$\left\{ \int (Q_t^*(s, \pi^*(s)) - Q_t^*(s, \hat\pi_{\hat\tau}))^2 \mathbb{I}[s \in \mathcal{S}_*] ds \right\}^{1/2} = O\left(\varepsilon^{1+\alpha}\right).$$

For the second term,

$$\left\{ \int (Q_t^*(s, \pi^*(s)) - Q_t^*(s, \hat\pi_{\hat\tau}))^2 \mathbb{I}\,[s \in \mathcal{S}_\epsilon^c] \, ds \right\}^{1/2} \leq \left\{ \int \left( \frac{4 \left| \hat\tau^{\hat\pi_{t+1}}(s) - \tau^{\pi_{t+1}^*, \circ}(s) \right|^2}{Q_t^*(s, \pi^*(s)) - Q_t^*(s, \hat\pi_{\hat\tau})} \right)^2 \mathbb{I}\,[s \in \mathcal{S}_\epsilon^c] \, ds \right\}^{1/2}$$

$$\leq \frac{4}{\varepsilon} \{ \int \left| \hat\tau^{\hat\pi_{t+1}}(s) - \tau^{\pi_{t+1}^*, \circ}(s) \right|^4 ds \}^{1/2} = O\left(\varepsilon^{-1} |\mathcal{I}|^{-2b_*}\right).$$

The result follows as previous. $\qquad\square$

*Proof of Theorem 1.* In the following, at times we omit the fixed evaluation policy $\pi^e$ from the notation for brevity. That is, in this proof, $\hat\tau_t, \tau_t^n$ are equivalent to $\hat\tau_t^{\pi^e}, \tau_t^{n, \pi^e}$. Further define

$$\nu_t = \hat\tau_t - \tau_t^n, \nu_t^\circ = \hat\tau_t - \tau_t^\circ$$

Strong convexity of the squared loss implies that:

$$D_{\tau_t, \tau_t} \mathcal{L}\left(\tau_t, \hat\eta\right) [\nu_t, \nu_t] \geq \lambda \|\nu_t\|_2^2$$

therefore

$$\frac{\lambda}{2}\|\nu_t\|_2^2 \leq \mathcal{L}_D(\hat{\tau}_t, \hat{\eta}) - \mathcal{L}_D(\tau_t^n, \hat{\eta}) - D_{\tau_t}\mathcal{L}_D(\tau_t^n, \hat{\eta})[\nu_t] \tag{17}$$

$$\leq \epsilon(\hat{\tau}_t, \hat{\eta}) - D_{\tau_t}\mathcal{L}_D(\tau_t^n, \eta^\circ)[\nu_t]$$
$$+ D_{\tau_t}\mathcal{L}_D(\tau_t^n, \eta^\circ)[\nu_t] - D_{\tau_t}\mathcal{L}_D(\tau_t^n, \hat{\eta})[\nu_t]$$

We bound each term in turn.

To bound $|D_{\tau_t}\mathcal{L}_D(\tau_t^n, \eta^\circ)[\nu_t]|$, note that

$$D_{\tau_t}\mathcal{L}_D(\tau_t^n, \eta^\circ)[\nu_t] = \mathbb{E}[(R + \gamma Q_{t+1} - V_t^{\pi^b, \pi_{t+1:T}} + (A - \pi_t^b)\tau_t))(A - \pi_t^b)\nu_t]$$

and by the properties of the conditional moment at the true $\tau^\circ$,

$$= \mathbb{E}[(R + \gamma Q_{t+1} - V_t^{\pi^b, \pi_{t+1:T}} + (A - \pi_t^b)\tau_t^\circ))(A - \pi_t^b)\nu_t] = 0$$

Therefore,

$$D_{\tau_t}\mathcal{L}_D(\tau_t^n, \eta^\circ)[\nu_t] = -\mathbb{E}[(\tau^\circ - \tau_t^n)(A - \pi_t^b)(A - \pi_t^b)(\hat{\tau}_t - \tau_t^n)]$$

Note that in general, for generic $p, q, r$ such that $1/p + 1/q + 1/r = 1$ we have that $\mathbb{E}[fgh] \leq \|fg\|_{p'}\|h\|_r \leq \|f\|_p\|g\|_q\|h\|_r$ where $p' = \frac{pq}{p+q}$ or $\frac{1}{p'} = \frac{1}{p} + \frac{1}{q}$ or $1 = \frac{1}{p/p'} + \frac{1}{q/p'}$.

Therefore,

$$D_{\tau_t}\mathcal{L}_D(\tau_t^n, \eta^\circ)[\nu_t] \leq |D_{\tau_t}\mathcal{L}_D(\tau_t^n, \eta^\circ)[\nu_t]|$$
$$\leq \mathbb{E}[(\tau^\circ - \tau_t^n)\mathbb{E}[(A_t - \pi_t^b)(A_t - \pi_t^b) \mid S_t](\hat{\tau}_t - \tau_t^n)]$$
$$\leq \|(\tau^\circ - \tau_t^n)\|_u\|(\hat{\tau}_t - \tau_t^n)\|_{\overline{u}} \cdot \left\{\sup_s \mathbb{E}[(A_t - \pi_t^b)(A_t - \pi_t^b) \mid s]\right\}$$

where $u, \overline{u}$ satisfy $\frac{1}{u} + \frac{1}{\overline{u}} = 1$.

Next we bound $D_{\tau_t}\mathcal{L}_D(\tau_t^n, \eta^\circ)[\nu_t] - D_{\tau_t}\mathcal{L}_D(\tau_t^n, \hat{\eta})[\nu_t]$ by universal orthogonality. By a second order Taylor expansion, we have that, where $\eta_\epsilon = \eta^\circ + \epsilon(\hat{\eta} - \eta^\circ)$.

$$D_{\tau_t}\left(\mathcal{L}_D(\tau_t^n, \eta^\circ) - \mathcal{L}_D(\tau_t^n, \hat{\eta})\right)[\nu_t] = \frac{1}{2}\int_0^1 D_{\eta, \eta, \tau_t}(\tau_t^n, \tau_{t+1}^\circ, \eta_\epsilon)[\hat{\eta} - \eta^\circ, \hat{\eta} - \eta^\circ, \nu_t]$$

We can deduce from Lemmas 4 and 5 that the integrand is:

$$\mathbb{E}\left[\tau_t^2\left(\hat{\pi}_t^b - \pi_t^{b,\circ}\right)^2\nu_t\right] + \mathbb{E}\left[(\hat{\pi}_t^b - \pi_t^{b,\circ})\tau_t(\hat{m}_t - m_t^\circ)\nu_t\right] + \mathbb{E}\left[(\hat{\pi}_t^b - \pi_t^{b,\circ})\tau_t\gamma(\hat{Q}_{t+1} - Q_{t+1}^\circ)\nu_t\right]$$
$$- \mathbb{E}\left[(\hat{m}_t - m_t^\circ)\gamma\left(\hat{Q}_{t+1} - Q_{t+1}^\circ\right)\nu_t\right]$$
$$\leq B_\tau^2\|\left(\hat{\pi}_t^b - \pi_t^{b,\circ}\right)^2\|_u\|\nu_t\|_{\overline{u}} + B_\tau\|(\hat{\pi}_t^b - \pi_t^{b,\circ})(\hat{m}_t - m_t^\circ)\|_u\|\nu_t\|_{\overline{u}} + \gamma B_\tau\|(\hat{\pi}_t^b - \pi_t^{b,\circ})(\hat{Q}_{t+1} - Q_{t+1}^\circ)\|_u\|\nu_t\|_{\overline{u}}$$
$$+ \gamma\|(\hat{m}_t - m_t^\circ)(\hat{Q}_{t+1} - Q_{t+1}^\circ)\|_u\|\nu_t\|_{\overline{u}}$$

Putting the bounds together, we obtain:

$$\frac{\lambda}{2}\|\nu_t\|_2^2 \leq \epsilon(\hat{\tau}_t, \hat{\eta}) + \|\nu_t\|_{\overline{u}}\|(\tau^\circ - \tau_t^n)\|_u$$
$$+ \|\nu_t\|_{\overline{u}}\left(B_\tau^2\|\left(\hat{\pi}_t^b - \pi_t^{b,\circ}\right)^2\|_u + B_\tau\|(\hat{\pi}_t^b - \pi_t^{b,\circ})(\hat{m}_t - m_t^\circ)\|_u + \gamma B_\tau\|(\hat{\pi}_t^b - \pi_t^{b,\circ})(\hat{Q}_{t+1} - Q_{t+1}^\circ)\|_u\right.$$
$$\left. + \gamma\|(\hat{m}_t - m_t^\circ)(\hat{Q}_{t+1} - Q_{t+1}^\circ)\|_u\right) \tag{18}$$

Let $\rho_t^{\pi^e}(\hat{\eta})$ denote the collected product error terms, e.g.

$$\rho_t^{\pi^e}(\hat{\eta}) = B_\tau{}^2 \|\left(\hat{\pi}_t^b - \pi_t^{b,\circ}\right)^2\|_u + B_\tau \|(\hat{\pi}_t^b - \pi_t^{b,\circ})(\hat{m}_t - m_t^\circ)\|_u$$
$$+ \gamma(B_\tau \|(\hat{\pi}_t^b - \pi_t^{b,\circ})(\hat{Q}_{t+1} - Q_{t+1}^\circ)\|_u + \|(\hat{m}_t - m_t^\circ)(\hat{Q}_{t+1} - Q_{t+1}^\circ)\|_u)$$

Analogously we drop the $\pi^e$ decoration from $\rho_t$ in this proof. The AM-GM inequality implies that for $x, y \geq 0, \sigma > 0$, we have that $xy \leq \frac{1}{2}(\frac{2}{\sigma}x^2 + \frac{\sigma}{2}y^2)$. Therefore

$$\frac{\lambda}{2}\|\nu_t\|_2^2 - \frac{\sigma}{4}\|\nu_t\|_{\bar{u}}^2 \leq \epsilon(\hat{\tau}_t, \hat{\eta}) + \frac{1}{\sigma}\left(\|(\tau^\circ - \tau_t^n)\|_u + \rho_t(\hat{\eta})\right)^2 \tag{19}$$

and since $(x+y)^2 \leq 2(x^2 + y^2)$,

$$\frac{\lambda}{2}\|\nu_t\|_2^2 - \frac{\sigma}{4}\|\nu_t\|_{\bar{u}}^2 \leq \epsilon(\hat{\tau}_t, \hat{\eta}) + \frac{2}{\sigma}\left(\|(\tau^\circ - \tau_t^n)\|_u^2 + \rho_t(\hat{\eta})^2\right)$$

$\square$

*Proof of Theorem 2.* Let $\hat{\mathcal{L}}_{S,t}, \hat{\mathcal{L}}_{S',t}$ denote the empirical loss over the samples in $S$ and $S'$; analogously $\hat{\eta}_S, \hat{\eta}_{S'}$ are the nuisance functions trained on each sample split.

Define the loss function $\ell_t$ on observation $O = \{(S_t, A_t, R_t, S_{t+1})\}_{t=1}^T$:

$$\ell_t(O; \tau_t; \hat{\eta}) = \left(\{R_t + \hat{Q}_{t+1}^{\pi_{t+1}^e}(S_{t+1}, A_{t+1}) - \hat{m}_t(S_t)\} - \{A - \hat{\pi}_t^b(1 \mid S_t)\} \cdot \tau_t(S_t)\right)^2$$

and the centered loss function $\Delta\ell$, centered with respect to $\hat{\tau}_t^n$:

$$\Delta\ell_t(O; \tau_t; \hat{\eta}) = \ell_t(O; \tau_t; \hat{\eta}) - \ell_t(O; \hat{\tau}_t^n; \hat{\eta}).$$

Assuming boundedness, $\ell_t$ is $L-$Lipschitz constant in $\tau_t$:

$$|\Delta\ell_t(O; \tau_t; \hat{\eta}) - \Delta\ell_t(O; \tau_t'; \hat{\eta})| \leq L\|\tau - \tau_t\|_2.$$

Note that $\ell(O, \hat{\tau}_t^n, \hat{\eta}) = 0$. Define the centered average losses:

$$\Delta\hat{\mathcal{L}}_{S,t}(\tau_t, \hat{\eta}) = \hat{\mathcal{L}}_{S,t}(\tau_t, \hat{\eta}) - \hat{\mathcal{L}}_{S,t}(\hat{\tau}_t^n, \hat{\eta}) = \hat{\mathbb{E}}_{n/2}^S[\Delta\ell_t(O, \tau_T, \hat{\eta})]$$
$$\Delta\mathcal{L}_{S,t}(\tau_t, \hat{\eta}) = \mathcal{L}_{S,t}(\tau_t, \hat{\eta}) - \mathcal{L}_{S,t}(\hat{\tau}_t^n, \hat{\eta}) = \mathbb{E}[\Delta\ell_t(O, \tau_T, \hat{\eta})]$$

Assume that $\delta_n$ is an upper bound on the critical radius of the centered function class $\{\Psi_{t,i}^n - \hat{\tau}_{t,i}^n$, with $\delta_n = \Omega(\frac{r \log\log n}{n})$, and define $\delta_{n,\xi} = \delta_n + c_0\sqrt{\frac{\log(c_1 T/\xi)}{n}}$ for some $c_0, c_1$.

By Lemma 7 (Lemma 14 of (Foster & Syrgkanis, 2019) on local Rademacher complexity decompositions), with high probability $1-\xi$, for all $t \in [T]$, and for $c_0$ a universal constant $\geq 1$.

$$|\Delta\mathcal{L}_{S,t}(\hat{\tau}_t, \hat{\eta}_{S'}) - \Delta\mathcal{L}_{D,t}(\hat{\tau}_t, \hat{\eta}_{S'})| = |\Delta\mathcal{L}_{S,t}(\hat{\tau}_t, \hat{\eta}_{S'}) - \Delta\mathcal{L}_{S,t}(\hat{\tau}_t^n, \hat{\eta}_{S'}) - (\Delta\mathcal{L}_{D,t}(\hat{\tau}_t, \hat{\eta}_{S'}) - \Delta\mathcal{L}_{D,t}(\hat{\tau}_t^n, \hat{\eta}_{S'}))|$$
$$\leq c_0\left(rm\delta_{n/2,\xi}\|\hat{\tau}_t - \hat{\tau}_t^n\|_2^2 + rm\delta_{n/2,\xi}^2\right)$$

Assuming realizability of $\hat{\tau}_t$, we have that $\frac{1}{2}\left(\Delta\hat{\mathcal{L}}_{S,t}(\hat{\tau}_t, \hat{\eta}_{S'}) + \Delta\hat{\mathcal{L}}_{S',t}(\hat{\tau}_t, \hat{\eta}_S)\right) \leq 0$. Then with high probability $\geq 1 - 2\xi$:

$$\frac{1}{2}\left(\Delta\mathcal{L}_{D,t}(\hat{\tau}_t, \hat{\eta}_{S'}) + \Delta\mathcal{L}_{D,t}(\hat{\tau}_t, \hat{\eta}_S)\right)$$
$$\leq \frac{1}{2}|\Delta\mathcal{L}_{D,t}(\hat{\tau}_t, \hat{\eta}_{S'}) - \Delta\mathcal{L}_{S,t}(\hat{\tau}_t, \hat{\eta}_{S'}) + \Delta\mathcal{L}_{D,t}(\hat{\tau}_t, \hat{\eta}_S) - \Delta\mathcal{L}_{S',t}(\hat{\tau}_t, \hat{\eta}_S)|$$
$$\leq \frac{1}{2}|\Delta\mathcal{L}_{D,t}(\hat{\tau}_t, \hat{\eta}_{S'}) - \Delta\mathcal{L}_{S,t}(\hat{\tau}_t, \hat{\eta}_{S'})| + |\Delta\mathcal{L}_{D,t}(\hat{\tau}_t, \hat{\eta}_S) - \Delta\mathcal{L}_{S',t}(\hat{\tau}_t, \hat{\eta}_S)|$$
$$\leq c_0\left(rm\delta_{n/2,\xi}\|\hat{\tau}_t - \hat{\tau}_t^n\|_2 + rm\delta_{n/2,\xi}^2\right)$$

The $\epsilon$ excess risk term in Theorem 1 indeed corresponds to one of the loss differences defined here, i.e. $\Delta\mathcal{L}_{D,t}(\hat{\tau}_t, \hat{\eta}_S) := \epsilon(\hat{\tau}_t^n, \hat{\tau}_t, \hat{h}_S)$. Therefore, applying Theorem 1 with $u = \bar{u} = 2$ and $\sigma = \lambda$ with the above bound, and averaging the sample-split estimators, we obtain

$$\frac{\lambda}{4}\|\nu_t\|_2^2 \leq \frac{1}{2}\left(\epsilon(\hat{\tau}_t, \hat{\eta}_S) + \epsilon(\hat{\tau}_t, \hat{\eta}_{S'})\right) + \frac{2}{\lambda}\left(\|\tau_t^\circ - \hat{\tau}_t^n\|_2^2 + \sum_{s\in\{S,S'\}}\rho_t(\hat{\eta}_s)^2\right)$$

We further decompose the excess risk of empirically-optimal $\hat{\tau}_t$ relative to the population minimizer to instead bound by the error of $\hat{\tau}_t$ to the projection onto $\Psi$, $\hat{\tau}_t^n$, since $\|\hat{\tau}_t - \tau_t^\circ\|_2^2 \leq \|\hat{\tau}_t - \hat{\tau}_t^n\|_2^2 + \|\hat{\tau}_t^n - \tau_t^\circ\|_2^2$, we obtain

$$\frac{\lambda}{4}\|\hat{\tau}_t - \tau_t^\circ\|_2^2 \leq c_0\left(rm\delta_{n/2,\xi}\|\hat{\tau}_t - \hat{\tau}_t^n\|_2 + rm\delta_{n/2,\xi}^2\right) + \frac{8+\lambda^2}{4\lambda}\|\tau_t^\circ - \hat{\tau}_t^n\|_2^2 + \frac{2}{\lambda}\sum_{s\in\{S,S'\}}\rho_t(\hat{\eta}_s)^2$$

Again using the AM-GM inequality $xy \leq \frac{1}{2}\left(\frac{2}{\sigma}x^2 + \frac{\sigma}{2}y^2\right)$, we bound

$$c_0\left(rm\delta_{n/2,\xi}\|\hat{\tau}_t - \hat{\tau}_t^n\|_2 + rm\delta_{n/2,\xi}^2\right) \leq \frac{c_0}{2}r^2m^2(1+\frac{2}{\epsilon})\delta_{n/2,\xi}^2 + \frac{\epsilon}{4}\|\hat{\tau}_t - \hat{\tau}_t^n\|_2^2$$

$$\leq c_0 r^2 m^2(1+\frac{1}{\epsilon})\delta_{n/2,\xi}^2 + \frac{\epsilon}{4}(\|\hat{\tau}_t - \tau_t^\circ\|_2^2 + \|\tau_t^\circ - \hat{\tau}_t^n\|_2^2)$$

Therefore,

$$\frac{\lambda-\epsilon}{4}\|\hat{\tau}_t - \tau_t^\circ\|_2^2 \leq c_0 r^2 m^2(1+\frac{1}{\epsilon})\delta_{n/2,\xi}^2 + \left(\frac{8+\lambda^2}{4\lambda} + \frac{\epsilon}{4}\right)\|\tau_t^\circ - \hat{\tau}_t^n\|_2^2 + \frac{2}{\lambda}\sum_{s\in\{S,S'\}}\rho_t(\hat{\eta}_s)^2$$

Choose $\epsilon \leq \lambda/8$ so that

$$\frac{\lambda}{8}\|\hat{\tau}_t - \tau_t^\circ\|_2^2 \leq c_0 r^2 m^2(1+\frac{8}{\lambda})\delta_{n/2,\xi}^2 + \left(\frac{4+\lambda^2}{2\lambda}\right)\|\tau_t^\circ - \hat{\tau}_t^n\|_2^2 + \frac{2}{\lambda}\sum_{s\in\{S,S'\}}\rho_t(\hat{\eta}_s)^2$$

$$\leq \left(1 + \frac{8}{\lambda} + \frac{\lambda}{2}\right)(c_0 r^2 m^2 \delta_{n/2,\xi}^2 + \|\tau_t^\circ - \hat{\tau}_t^n\|_2^2 + \sum_{s\in\{S,S'\}}\rho_t(\hat{\eta}_s)^2)$$

and therefore

$$\|\hat{\tau}_t - \tau_t^\circ\|_2^2 \leq \left(\frac{8}{\lambda}(1+\frac{8}{\lambda}) + 4\right)(c_0 r^2 m^2 \delta_{n/2,\xi}^2 + \|\tau_t^\circ - \hat{\tau}_t^n\|_2^2 + \sum_{s\in\{S,S'\}}\rho_t(\hat{\eta}_s)^2)$$

Taking expectations:

$$\mathbb{E}[\|\hat{\tau}_t - \tau_t^\circ\|_2^2] \leq \left(\frac{8}{\lambda}(1+\frac{8}{\lambda}) + 4\right)(c_0 r^2 m^2 \delta_{n/2}^2 + \|\tau_t^\circ - \hat{\tau}_t^n\|_2^2 + \max_{s\in\{S,S'\}}\mathbb{E}[\rho_t(\hat{\eta}_s)^2])$$

Therefore, if the product error rate terms are all of the same order as the estimation order terms:

$$\mathbb{E}[\|\hat{\pi}_t^b - \pi_t^{b,\circ}\|_2^2] = O(\delta_{n/2}^2 + \|\tau_t^\circ - \hat{\tau}_t^n\|_2^2)$$

$$\mathbb{E}[\|(\hat{\pi}_t^b - \pi_t^{b,\circ})(\hat{m}_t - m_t^\circ)\|_2^2] = O(\delta_{n/2}^2 + \|\tau_t^\circ - \hat{\tau}_t^n\|_2^2)$$

$$\mathbb{E}[\|(\hat{\pi}_t^b - \pi_t^{b,\circ})(\hat{Q}_{t+1} - Q_{t+1}^\circ)\|_2^2] = O(\delta_{n/2}^2 + \|\tau_t^\circ - \hat{\tau}_t^n\|_2^2)$$

$$\mathbb{E}[\|(\hat{m}_t - m_t^\circ)(\hat{Q}_{t+1} - Q_{t+1}^\circ)\|_2^2] = O(\delta_{n/2}^2 + \|\tau_t^\circ - \hat{\tau}_t^n\|_2^2)$$

$\square$

Full statement of Theorem 3:

**Theorem 4** (Policy optimization bound, full statement of Theorem 3). *Suppose Assumptions 1 to 4. Further, suppose that $Q^\circ$ satisfies Assumption 7 (margin) with $\alpha > 0$. Suppose the product error rate conditions hold for each $t$ for data-optimal policies evaluated along the algorithm, i.e. for each $t$, for $\hat{\underline{\pi}}_{t+1}$, each of $\mathbb{E}[\|(\hat{\pi}_t^b - \pi_t^{b,\circ})\|_2^2]$, $\mathbb{E}[\|(\hat{\pi}_t^b - \pi_t^{b,\circ})(\hat{m}_t^{\hat{\underline{\pi}}_{t+1}} - m_t^{\circ,\hat{\underline{\pi}}_{t+1}})\|_2^2]$, $\mathbb{E}[\|(\hat{\pi}_t^b - \pi_t^{b,\circ})(\hat{Q}_{t+1}^{\hat{\underline{\pi}}_{t+2}} - Q_{t+1}^{\circ,\hat{\underline{\pi}}_{t+2}})\|_2^2]$, and $\mathbb{E}[\|(\hat{m}_t - m_t^\circ)(\hat{Q}_{t+1}^{\circ,\hat{\underline{\pi}}_{t+2}} - Q_{t+1}^{\circ,\hat{\underline{\pi}}_{t+2}})\|_2^2]$ are of order $O(\delta_{n/2}^2 + \|\tau_t^{\hat{\pi}_{t+1},\circ} - \tau_t^{\hat{\pi}_{t+1},n}\|_2^2)$. Suppose that then for $\hat{\pi}_t$, Theorem 2 holds, and the critical radius $\delta_{n/2}$ and for time $t$, function class specification error $\|\tau_t^{\hat{\pi}_{t+1},\circ} - \tau_t^{\hat{\pi}_{t+1},n}\|_2$ satisfy the root-mean-squared-error rate conditions. That is, for rates $\rho_t^{(c)}, \rho_t^{(\Psi)}$, we have that $\delta_{n/2}^2 = K_r^2 n^{-2\rho_t^{(c)}}$, $\|\tau_t^{\hat{\pi}_{t+1},\circ} - \tau_t^{\hat{\pi}_{t+1},n}\|_2^2 = K_\Psi^2 n^{-2\rho_t^{(\Psi)}}$. Define for a generic $t$, the slowest rate after $t$ as $\bar{\rho}_{\geq t}^{(\cdot)} = \min_{t' \geq t}\{\rho_{t'}^{(\cdot)}\}$. Then, for $(\cdot) \in \{(c),(\Psi)\}$. Then, with high probability $\geq n^{-\kappa}, \|\hat{\tau}_t^{\hat{\pi}_{t+1}} - \tau_t^{\circ,\underline{\pi}_{t+1}^*}\| \leq O(\delta_{n/2} + \|\tau_t^{\circ,\hat{\underline{\pi}}_{t+1}} - \tau_t^{n,\hat{\underline{\pi}}_{t+1}}\|_2) + Kn^{-\mathcal{R}_t}$. where*

$$\mathcal{R}_k = \min\left(\rho_{k+1}^{(c)} \cdot \frac{2+2\alpha}{2+\alpha}, \ \rho_{k+1}^{(\Psi)} \cdot \frac{2+2\alpha}{2+\alpha}, \ \left\{\min_{k' \geq k+1}(\rho_{k'}^{(c)}, \rho_{k'}^{(\Psi)})\right\} \cdot \frac{2+2\alpha^{T-k'}}{2+\alpha}\right).$$

*Further suppose that for $t' \geq t$, we have that $\rho_t^{(\cdot)} \leq \rho_{t'}^{(\cdot)}$, for $(\cdot) \in \{(c),(\Psi)\}$, i.e. the estimation error rate is nonincreasing over time. Then,*

$$\|\hat{\tau}_t^{\hat{\pi}_{t+1}} - \tau_t^{\circ,\underline{\pi}_{t+1}^*}\|_2 \leq O(\delta_{n/2} + \|\tau_t^{\circ,\hat{\underline{\pi}}_{t+1}} - \tau_t^{n,\hat{\underline{\pi}}_{t+1}}\|_2), \tag{20}$$

*and*

$$\left|\mathbb{E}[V_1^{\pi^*}(S_1) - V_1^{\hat{\pi}_{\hat{\tau}}}(S_1)]\right| = O(n^{-\min\{\bar{\rho}_{\geq 1}^{(c)}, \bar{\rho}_{\geq 1}^{(\Psi)}\}\frac{2+2\alpha}{2+\alpha}}).$$

*Proof of Theorem 3.* **Preliminaries** We introduce some additional notation. For the analysis of implications of policy optimization, we further introduce notation that parametrizes the time-$t$ loss function with respect to the time-$(t+1)$ policy. In analyzing the policy optimization, this will be used to decompose the policy error arising from time steps closer to the horizon. Define

$$\mathcal{L}_D(\tau_t^n, \tau_{t+1}', \hat{\eta}) = \mathbb{E}\left[\left(\{R_t + \gamma Q_{t+1}^{\pi_{\tau_{t+1}'}}(S_{t+1}, A_{t+1}) - V_{\pi_t^b, \pi_{\tau_{t+1}'}}(S_t)\} - \{A - \pi_t^b(1 \mid S_t)\} \cdot \tau(S_t)\right)^2\right]$$

where $\pi_{\tau_{t+1}'}(s) \in \arg\max \tau_{t+1}'(s)$. That is, the second argument parameterizes the difference-of-$Q$ function that generates the policy that oracle nuisance functions are evaluated at.

Then, for example, the true optimal policy satisfies that $\pi_t^* \in \arg\max \tau_t^\circ(s)$. We define the oracle loss function with nuisance functions evaluated with respect to the optimal policy $\pi^*$.

$$\mathcal{L}_D(\tau_t^n, \tau^\circ, \hat{\eta}) = \mathbb{E}\left[\left(\{R_t + \gamma Q_{t+1}^{\pi_{\tau_{t+1}^*}}(S_{t+1}, A_{t+1}) - m^\circ(S_t)\} - \gamma\{A - \pi_t^b(1 \mid S_t)\} \cdot \tau(S_t)\right)^2\right]$$

In contrast, the empirical policy optimizes with respect to a next-stage *estimate* of the *empirical best* next-stage policy $\hat{\pi}_{\hat{\tau}_{t+1}}$. That is, noting the empirical loss function:

$$\mathcal{L}_D(\tau_t^n, \hat{\tau}_{t+1}, \hat{\eta}) = \mathbb{E}\left[\left(\{R_t + \gamma Q_{t+1}^{\hat{\pi}_{\hat{\tau}_{t+1}}}(S_{t+1}, A_{t+1}) - m^\circ(S_t)\} - \gamma\{A - \pi_t^b(1 \mid S_t)\} \cdot \tau(S_t)\right)^2\right]$$

**Step 1: Applying advantage estimation results.** At every timestep, the first substep is to estimate the $Q$-function contrast, $\hat{\tau}_t^{\hat{\pi}_{t+1}}$. The assumptions on product error nuisance rates imply that for a fixed $\hat{\pi}_{t+1}$ that we would obtain estimation error

$$\mathbb{E}\left[\|\hat{\tau}_t^{\hat{\pi}_{t+1}} - \tau_t^{\hat{\pi}_{t+1},\circ}\|_2^2\right] = O\left(\delta_{n/2}^2 + \left\|\tau_t^{\pi^e,\circ} - \tau_t^{\pi^e,n}\right\|_2^2\right)$$

**Step 2: Establishing policy consistency.** Applying Lemma 2 requires a convergence rate of $\hat{\tau}_t^{\hat{\pi}_{t+1}}$ to $\hat{\tau}_t^{\pi^*_{t+1}}$. The estimation error guarantees on the contrast function, however, are for the policy $\hat{\pi}_{t+1}$. We obtain the required bound via induction. At a high level, the estimation error arising from $\hat{\pi}_{t+1}$ vs $\pi^*_{t+1}$ too eventually is integrated; so when the margin exponent $\alpha > 0$, these policy error terms are higher-order and vanish at a faster rate.

Importantly, we suppose the product error rate conditions hold for each $t$ for data-optimal policies evaluated along the algorithm, i.e. for each $t$, for each $t$, for $\hat{\underline{\pi}}_{t+1}$, each of $\mathbb{E}[\|(\hat{\pi}_t^b - \pi_t^{b,\circ})\|_2^2]$, $\mathbb{E}[\|(\hat{\pi}_t^b - \pi_t^{b,\circ})(\hat{m}_t^{\hat{\underline{\pi}}_{t+1}} - m_t^{\circ,\hat{\underline{\pi}}_{t+1}})\|_2^2]$, $\mathbb{E}[\|(\hat{\pi}_t^b - \pi_t^{b,\circ})(\hat{Q}_{t+1}^{\hat{\underline{\pi}}_{t+2}} - Q_{t+1}^{\circ,\hat{\underline{\pi}}_{t+2}})\|_2^2]$, and $\mathbb{E}[\|(\hat{m}_t - m_t^\circ)(\hat{Q}_{t+1}^{\circ,\hat{\underline{\pi}}_{t+2}} - Q_{t+1}^{\circ,\hat{\underline{\pi}}_{t+2}})\|_2^2]$ are of order $O(\delta_{n/2}^2 + \|\tau_t^{\hat{\pi}_{t+1},\circ} - \tau_t^{\hat{\pi}_{t+1},n}\|_2^2)$.

**Step 2a**: induction hypothesis.

Next we show the induction hypothesis.

First we consider the base case: When $t = T$, $\tau_T$ is independent of the forward policy so that $\|\hat{\tau}_T^{\hat{\pi}} - \tau_T^{\circ,\pi^*}\| = \|\hat{\tau}_T - \tau_T^\circ\|$. Then the base case follows by Theorem 2.

Suppose it is true that for timesteps $k \geq t + 1$, we have that

$$\|\hat{\tau}_k^{\hat{\underline{\pi}}_{k+1}} - \tau_k^{\circ,\underline{\pi}^*_{k+1}}\| = O(\delta_{n/2} + \|\tau_k^{\circ,\hat{\underline{\pi}}_{k+1}} - \tau_k^{n,\hat{\underline{\pi}}_{k+1}}\|_2) + Kn^{-\mathcal{R}_k}, \tag{21}$$

where

$$\mathcal{R}_k = \min\left(\rho_{k+1}^{(c)} \cdot \frac{2+2\alpha}{2+\alpha}, \ \rho_{k+1}^{(\Psi)} \cdot \frac{2+2\alpha}{2+\alpha}, \ -\{\min_{k'\geq k+1}(\rho_{k'}^{(c)}, \rho_{k'}^{(\Psi)})\} \cdot \frac{2+2\alpha}{2+\alpha}^{T-k'}\right). \tag{22}$$

And therefore, applying Lemma 2, that

$$\left|\mathbb{E}[V_k^{\pi^*} - V_k^{\hat{\pi}_{\hat{\tau}}}]\right| = O(n^{-\min\{\rho_k^{(c)},\rho_k^{(\Psi)}\}\frac{2+2\alpha}{2+\alpha}}) + o(n^{-\min\{\rho_k^{(c)},\rho_k^{(\Psi)}\}\frac{2+2\alpha}{2+\alpha}}). \tag{23}$$

We will show that the induction hypothesis implies

$$\|\hat{\tau}_k^{\hat{\underline{\pi}}_{t+1}} - \tau_k^{\circ,\underline{\pi}^*_{t+1}}\| \leq O(\delta_{n/2} + \|\tau_k^{\circ,\hat{\underline{\pi}}_{t+1}} - \tau_t^{n,\hat{\underline{\pi}}_{t+1}}\|_2) + Kn^{-\mathcal{R}_t}.$$

and

$$\left|\mathbb{E}[V_k^{\pi^*} - V_k^{\hat{\pi}_{\hat{\tau}}}]\right| = O(n^{-\min\{\rho_k^{(c)},\rho_k^{(\Psi)}\}\frac{2+2\alpha}{2+\alpha}}) + o(n^{-\min\{\rho_k^{(c)},\rho_k^{(\Psi)}\}\frac{2+2\alpha}{2+\alpha}})$$

First decompose the desired error $\|\hat{\tau}_k^{\hat{\underline{\pi}}_{t+1}} - \tau_k^{\circ,\underline{\pi}^*_{t+1}}\|$ as:

$$\|\hat{\tau}_k^{\hat{\underline{\pi}}_{t+1}} - \tau_k^{\circ,\underline{\pi}^*_{t+1}}\| \leq \|\hat{\tau}_k^{\hat{\underline{\pi}}_{t+1}} - \tau_k^{\circ,\hat{\underline{\pi}}_{t+1}}\| + \|\tau_k^{\circ,\hat{\underline{\pi}}_{t+1}} - \tau_k^{\circ,\underline{\pi}^*_{t+1}}\| \tag{24}$$

The first term is the policy evaluation estimation error, and under the product error rate assumptions , Theorems 1 and 2 give that $\mathbb{E}[\|\hat{\tau}_k^{\hat{\underline{\pi}}_{t+1}} - \tau_k^{\circ,\hat{\underline{\pi}}_{t+1}}\|_2^2] = O(\delta_{n/2}^2 + \|\tau_k^{\circ,\hat{\underline{\pi}}_{t+1}} - \tau_k^{n,\hat{\underline{\pi}}_{t+1}}\|_2^2)$. The second term of the above depends on the convergence of the empirically optimal policy $\hat{\pi}$; we use our analysis from Lemma 2 to bound the impact of future estimates of difference-of-$Q$ functions using the induction hypothesis. The following analysis will essentially reveal that the margin assumption of Assumption 7 implies that the error due to the empirically optimal policy is higher-order, and the first term (time$-t$ estimation error of $\hat{\tau}_k$) is the leading term.

As in eq. (9), we have that:

$$V_t^*(s) - V_t^{\pi_{\hat{\tau}}}(s) \leq \gamma\mathbb{E}_{\hat{\pi}_t}\left[V_{t+1}^{\pi^*} - V_{t+1}^{\hat{\pi}_{\hat{\tau}}} \mid s_t\right] + Q_t^*(s,\pi^*) - Q_t^*(s,\hat{\pi}_{\hat{\tau}}).$$

Decompose:

$$\|\tau_t^{\circ,\hat{\underline{\pi}}_{t+1}} - \tau_t^{\circ,\underline{\pi}^*_{t+1}}\| \leq \sum_a \|Q_t^{\underline{\pi}^*_{t+1}}(s,a) - Q_t^{\hat{\underline{\pi}}_{t+1}}(s,a)\|$$

By definition of $\tau$ and $\pm V_{t+1}^{\hat{\pi}_{t+1},\pi_{t+2}^*}$, for each $a$, we have that

$$\|Q_t^{\pi_{t+1}^*}(s,a) - Q_t^{\hat{\pi}_{t+1}}(s,a)\|$$

$$= \|\mathbb{E}_{\pi_t^a}[V_{t+1}^{\pi_{t+1}^*} - V_{t+1}^{\hat{\pi}_{t+1}} \mid S_t]\|$$

$$\leq \|\mathbb{E}_{\pi_t^a}[V_{t+1}^{\pi_{t+1}^*} - V_{t+1}^{\hat{\pi}_{t+1},\pi_{t+2}^*} \mid S_t]\| + \|\mathbb{E}_{\pi_t^a}[V_{t+1}^{\hat{\pi}_{t+1},\pi_{t+2}^*} - V_{t+1}^{\hat{\pi}_{t+1}} \mid S_t]\|$$

$$= \|\mathbb{E}_{\pi_t^a}[Q_{t+1}^{\pi_{t+2}^*}(S_{t+1},\pi_{t+1}^*) - Q_{t+1}^{\pi_{t+2}^*}(S_{t+1},\hat{\pi}_{t+1}) \mid S_t]\| + \gamma\|\mathbb{E}_{\pi_t^a}[\mathbb{E}_{\hat{\pi}_{t+1}}[V_{t+2}^{\pi_{t+2}^*} - V_{t+2}^{\hat{\pi}_{t+2}} \mid S_t]]\| \qquad (25)$$

$$\leq c\left\{\int (Q_{t+1}^{\pi_{t+2}^*}(s,\pi_{t+1}^*) - Q_{t+1}^{\pi_{t+2}^*}(s,\hat{\pi}_{t+1}))^2 ds\right\}^{1/2} + \gamma\|\mathbb{E}_{\pi_t^a}[\mathbb{E}_{\hat{\pi}_{t+1}}[V_{t+2}^{\pi_{t+2}^*} - V_{t+2}^{\hat{\pi}_{t+2}} \mid S_t]]\| \qquad (26)$$

where the last inequality follows by Assumption 4 and the policy-convolved transition density.

Next we bound the first term using the margin analysis of Lemma 2 and the inductive hypothesis. Supposing the product error rates are satisfied on the nuisance functions for estimation of $\hat{\tau}_{t+1}$, the induction hypothesis gives that

$$\mathbb{E}[\|\hat{\tau}_{t+1}^{\hat{\pi}_{t+2}} - \tau_{t+1}^{\circ,\pi_{t+2}^*}\|_2] = O\left(\delta_{n/2} + \|\tau_t^{\pi^e,\circ} - \tau_t^n\|_2 + n^{-\mathcal{R}_{t+1}}\right).$$

The induction hypothesis gives the integrated risk rate assumption on $\hat{\tau}_{t+1}$ to apply Lemma 2,

$$\left\{\int (Q_{t+1}^{\pi_{t+2}^*}(s,\pi_{t+1}^*) - Q_{t+1}^{\pi_{t+2}^*}(s,\hat{\pi}_{t+1}))^2 ds\right\}^{1/2}$$

$$\leq \frac{(1-\gamma^{T-t-1})}{1-\gamma} c(T-t-1)\{O(n^{-\kappa}) + Kn^{-\min\{r_{t+1}^{(c)}, r_{t+1}^{(\Psi)}, \mathcal{R}_{t+1}\}(1+\alpha)}\}.$$

Combining with the previous analysis, we obtain:

$$\|\hat{\tau}_t^{\hat{\pi}_{t+1}} - \tau_t^{\circ,\pi_{t+1}^*}\|_2^2 \leq O(\delta_{t,n/2}^2 + \|\tau_t^{\circ,\hat{\pi}_{t+1}} - \tau_t^{n,\hat{\pi}_{t+1}}\|_2^2) + O(n^{-\min\left\{\rho_{t+2}^{(c)}, \rho_{t+2}^{(\Psi)}, \mathcal{R}_{t+2}\right\}\frac{2+2\alpha}{2+\alpha}})\}$$

$$+ \frac{(1-\gamma^{T-t-1})}{1-\gamma} c(T-t-1)\{O(n^{-\kappa}) + Kn^{-\min\{\rho_{t+1}^{(c)}, \rho_{t+1}^{(\Psi)}, \mathcal{R}_{t+1}\}\frac{2+2\alpha}{2+\alpha}}\} \qquad (27)$$

from eq. (25) and appendix A.3.

Hence we obtain the inductive step and the result follows.

If we further assume that for $t' \geq t$, we have that $\rho_t^{(\cdot)} \leq \rho_{t'}^{(\cdot)}$, for $(\cdot) \in \{(c),(\Psi)\}$, i.e. the estimation error rate is nonincreasing over time, and that $\alpha > 0$ (i.e. Assumption 7, the margin assumption, holds with exponent $\alpha > 0$, then we can see from the result that the integrated risk terms obtain faster rates, hence are higher-order, and the leading term is the auxiliary estimation error of the $Q$-function contrast.

$\square$

# B RESULTS USED FROM OTHER WORKS

Here we collect technical lemmas from other works, stated without proof.

**Lemma 6** (Lemma 18 of (Lewis & Syrgkanis, 2021)). *Consider any sequence of non-negative numbers $a_1, \ldots, a_m$ satisfying the inequality:*

$$a_t \leq \mu_t + c_t \max_{j=t+1}^m a_j$$

*with $\mu_t, c_t \geq 0$. Let $c := \max_{t \in [m]} c_t$ and $\mu := \max_{t \in [m]} \mu_t$. Then it must also hold that:*

$$a_t \leq \mu \frac{c^{m-t+1} - 1}{c - 1}$$

**Lemma 7** (Lemma 14 of (Foster & Syrgkanis, 2019), see also results on local Rademacher complexity (Wainwright, 2019)). *Consider a function class $\mathcal{F}$, with $\sup_{f \in \mathcal{F}} \|f\|_\infty \leq 1$, and pick any $f^* \in \mathcal{F}$. Let $\delta_n^2 \geq \frac{4d \log(41 \log(2c_2 n))}{c_2 n}$ be any solution to the inequalities:*

$$\forall t \in \{1, \ldots, d\} : \mathcal{R}\left(\text{star}\left(\mathcal{F}|_t - f_t^*\right), \delta\right) \leq \delta^2.$$

*Moreover, assume that the loss $\ell$ is L-Lipschitz in its first argument with respect to the $\ell_2$ norm. Then for some universal constants $c_5, c_6$, with probability $1 - c_5 \exp\left(c_6 n \delta_n^2\right)$,*

$$\left|\mathbb{P}_n\left(\mathcal{L}_f - \mathcal{L}_{f^*}\right) - \mathbb{P}\left(\mathcal{L}_f - \mathcal{L}_{f^*}\right)\right| \leq 18 L d \delta_n \left\{\|f - f^*\|_2 + \delta_n\right\}, \quad \forall f \in \mathcal{F}.$$

*Hence, the outcome $\hat{f}$ of constrained ERM satisfies that with the same probability,*

$$\mathbb{P}\left(\mathcal{L}_{\hat{f}} - \mathcal{L}_{f^*}\right) \leq 18 L d \delta_n \left\{\left\|\hat{f} - f^*\right\|_2 + \delta_n\right\}.$$

*If the loss $\mathcal{L}_f$ is also linear in $f$, i.e. $\mathcal{L}_{f+f'} = \mathcal{L}_f + \mathcal{L}_{f'}$ and $\mathcal{L}_{\alpha f} = \alpha \mathcal{L}_f$, then the lower bound on $\delta_n^2$ is not required.*

## C   EXPERIMENTAL DETAILS

All experiments were ran either on a Macbook Pro M1 with 16gb RAM and 8 CPU cores or on a computer cluster with 64 CPU cores of 8gb RAM each. Experiments were run in Python using native Python, CVXPY, and scikit-learn. Each figure took approximately 3-10 minutes to generate.

**1d validation example (Section 4).** Following the specification of (Kallus & Uehara, 2019a, Sec 5.1), we consider a small MDP of $T = 30$, binary actions, univariate continuous state, initial state distribution $p(s_0) \sim \mathcal{N}(0.5, 0.2)$, transition probabilities $P_t(s_{t+1} \mid s_t, a_t) \sim \mathcal{N}(s + 0.3a - 0.15, 0.2)$. The target and behavior policies we consider are $\pi^e(a \mid s) \sim \text{Bernoulli}(p_e), p_e = 0.2/(1 + \exp(-0.1s)) + 0.2U, U \sim \text{Uniform}[0, 1]$ and $\pi^b(a \mid s) \sim \text{Bernoulli}(p_b), p_b = 0.9/(1 + \exp(-0.1s)) + 0.1U, U \sim \text{Uniform}[0, 1]$. We consider the interacted state-action basis, i.e. fit $Q$ on $s + s * a$ with an intercept. When $Q$ is well-specified, we do nearly exactly recover the right contrast function; although in such a small and well-specified example we do not see benefits of orthogonality.

**Extending to nonlinear settings: details on MINE.** Sample-based MI estimation methods with upper bounds (Cheng et al., 2020) and lower bounds (Belghazi et al., 2018b) provide an alternative solution by shuffling the sample from the joint distribution in the batch scope. In practice, we can use the statistics network to exploit the mutual information neural estimator (MINE) and optimize the absolute value in case of minor estimation errors. While mutual information is non-negative as a measure, the neural network estimates can be negative. Thus, we optimize the absolute value of the mutual information as our objective. This treats both positive and negative values consistently, driving the objective to zero and making the variables independent, as desired. Therefore, although we appreciate the concern that minimizing a function using a lower bound is less clear than maximizing it, our approach provides a pragmatic solution to the problem at hand, and our results confirm its effectiveness.

