# OpenReview forum: "Orthogonalized Estimation of Difference of Q-functions"
_ICLR.cc/2025/Conference — Submitted to ICLR 2025_

### Official Review · Reviewer_ynuS · 2024-10-21

**Soundness:** 3
**Presentation:** 3
**Contribution:** 3
**Rating:** 6
**Confidence:** 4

**Summary:**

The paper introduces a novel approach with statistical theory for offline reinforcement learning (RL) through the development of a dynamic generalization of the R-learner. This method is designed for estimating and optimizing the difference between Q-functions under two different actions for discrete-valued actions. The key contributions and results of this work are as follows:
The authors extend the R-learner framework, commonly used in causal inference, to the offline RL setting. This generalization allows for the estimation of Q-function differences, which can be utilized to optimize decision-making for multiple actions.

One of the contributions of this work is the use of orthogonal estimation techniques, which improves convergence rates and as a result mitigates the effects of unstable estimations in behavior policies, which is a common issue in offline RL.

The proposed method focuses on estimating a more structured Q-function contrast, which adapts to underlying sparse or smoother structures in the data. This is crucial because causal contrasts, such as the difference in outcomes between actions, often exhibit more structure (e.g., sparsity) than the Q-functions themselves. The method, therefore, leverages this property to improve policy optimization.

Theoretical guarantees for the proposed method are given, ensuring consistency in policy optimization. The framework is capable of estimating causal contrasts in a sequential decision-making setting, with applications for learning dynamic treatment rules based on observational data. Synthetic experiments are presented to demonstrate the effectiveness of their proposed approach.

**Strengths:**

Originality: This paper has originality. First, it extends the R-learner from causal inference to the dynamic setting of offline RL. I agree that focusing on Q-function contrasts rather than estimating full Q-functions is a novel approach, offering advantages in terms of adaptability to structured or smoother forms like sparsity. Additionally, the use of orthogonal estimation to stabilize policy optimization and improve convergence rates is an original contribution to offline RL, leveraging causal inference techniques in a new domain. The method's ability to adapt to sparse structures in the Q-function contrast further enhances its uniqueness, aligning it with modern trends in machine learning that exploit underlying structure for efficiency.

Quality: This paper has good quality. First, the application of orthogonal estimation techniques in this context seems a non-trivial approach to me, and it addresses critical challenges in offline RL such as slow convergence.

Clarity: This paper states its contributions clearly. In addition, as a theoretical work, the paper clearly introduced its mathematical setup: the assumptions and theorems are clearly specified. An illustration of the sequential decision making structure is given by figure 1.

Significance: This paper seeks to address a major problem in offline RL. Specifically, the instability of policy optimization due to slow convergence of nuisance functions (Q-functions and behavior policies). The incorporation of orthogonal estimation improves the stability and reliability of learning. The focus on estimating Q-function contrasts rather than the entire Q-function offers a more adaptable and potentially more efficient way to optimize decisions, especially in environments where the contrast between actions is more structured or smoother. This could lead to better policy performance in practice, addressing the need for more efficient learning in complex decision-making problems.

**Weaknesses:**

As a theoretical work, the paper does not clearly highlight the main technical challenges in its analysis. While the incorporation of orthogonal estimation to improve the robustness of offline RL is a valid contribution, it would be stronger if the paper explained why this is technically challenging. Alternatively, discussing the difficulties in the theoretical analysis of the proposed algorithm would enhance its depth.
Indeed, the paper claimed to have followed the analysis of Foster and Syrgkanis for orthogonal statistical learning. This is not an issue but the paper should also elaborate what makes the analysis in this paper different from the previous theoretical analysis, if applicable. For example, does the paper have a distinct error decomposition? Does the convergence analysis for Theorem 2 become different from previous work because some error term takes a different form? etc.

I think the following might be some helpful suggestion the authors could consider (if applicable).

(1) Include a paragraph after Theorem 2 or 3 explaining the key technical difficulties they encountered in proving those results.

(2) Explicitly compare their error decomposition or convergence analysis to previous work, pointing out any additional terms which that bring extra difficulty to the proof.

(3) Add a short paragraph at the beginning of the theoretical analysis session to highlight the key technical challenge and novelty on a high level.

In summary, the paper could benefit from more explicit discussion of the theoretical challenges or innovations in its analysis.

Minor issue:
There seems to be some typos in the sentence from line 419-422.

**Questions:**

Please discuss the unique theoretical challenge or innovations, as pointed out in **Weaknesses**.

---

> ### Author Response · Authors · 2024-11-29
> **response and clarifications**
>
> Thank you for your thorough review and recognizing the strengths of the paper, especially the fact that it doesn’t introduce dependence on unstable inverse behavior policy weights, which we think has hampered adoption of orthogonal ML or double ML estimation in offline RL in the past.
>
> We’d like to respond with clarifications to your concerns on weaknesses below. We have already included some of your great suggestions in the updated revision but have given them all careful consideration.
>
> Weaknesses:
> > does not clearly highlight …
> In lines 271-273 and lines 373-377 we have highlighted where our analysis is similar to prior work, and where the novelty and technical challenges arose.
>
> We wish to emphasize our work as a methodological contribution with rigorous theoretical guarantees. Our primary goal was to develop orthogonalized estimators that address statistical challenges and adapt to structure. The essence of our contribution lies in deriving a new estimand and orthogonal loss function.
>
> While our analysis process aligns with the high-level framework of orthogonal ML, what is new to our analysis is confirming that the functional derivative of our loss function wrt to the nuisance functions is 0, as shown in Lemmas 4-5. We will add this to the exposition of Thm. 2. This leads to a distinct error decomposition, although we believe the analysis remains fundamentally similar. The analysis framework for _all orthogonal statistical learning_ is similar: derive an estimator that is Neyman orthogonal to nuisance functions, i.e. its functional derivative wrt nuisance functions is 0. Then study the von Mises expansion of the estimator: establish that the second-order term is a higher-order remainder and leverage the error decomposition to conclude the nature of product-rate improvements. The last part appears in our new Assumption 5.
>
> To clarify our objectives, we aimed to create a new estimand and estimator to tackle specific problems, such as applying MI regularization in empirical RL on the difference-of-Q functions. We chose to modify the estimand as an elegant solution for adapting to sparse structures, rather than developing various methods for different structural assumptions, as illustrated in our experiments. This required us to develop and analyze a novel approach for estimating the difference-of-Q functions, which we believe is of independent interest.
>
> However, the challenges for our goals come up in establishing that orthogonal estimation implies convergent policy optimization, which is more difficult because each policy optimization step in principle defines a new set of nuisance functions. We do this with a mix of conceptual (our new estimator) and technical innovations (our induction analysis with margin) that are specific to our task. We explicitly designed our estimation framework to sidestep most of these challenges and moved this discussion of our conceptual innovations earlier in the related work, see lines 100-105 in revision, to make this clearer.

---

> > ### Author Response · Authors · 2024-11-29
> > **response, continued**
> >
> > > the exact distinction from Foster and Syrgkanis
> >
> > Perhaps it makes more sense to highlight the difference from Lewis and Syrgkanis 2021, since Foster and Syrgkanis was a very general paper providing a very general analysis framework for different estimands. Lewis and Syrgkanis studied a similar generalization of the R-learner, but they considered a non-Markovian dynamic treatment regime with only a terminal reward, and they did not estimate the analogue of the difference-of-Q-function but rather heterogeneity in non-time-varying covariates alone. Because their estimand was non-Markovian and fully sequential, there were additional nuisance functions. Because our estimand is different, our error decomposition resulting in orthogonality is different, though we fully believe the general recipe for orthogonality analysis is standard for double ML.
> >
> > Again, as mentioned in lines 373-377, our main technical challenge is proving policy optimization convergence, when we need to account for the estimation error arising because our nuisance estimation is based on estimates of the optimal policy: we want to show convergence of
> > $\hat{\tau}^{\hat{\pi}_{t+1}}_t$ for
> >
> > $\hat{\tau}^{\pi_{t+1}^*}_t$, although our policy evaluation analysis applies for the former.
> >
> > As mentioned, this requires leveraging the margin assumption to show that the estimation error arising from error in estimation of future-policies is higher-order.
> >
> >
> >
> > > Helpful suggestions:
> >
> > Thank you for these, we appreciate your suggestions. We have carefully considered them and completed the following revisions, in the attached version.
> >
> > (1) we already had a paragraph after Thm 3 in the original version.
> >
> > (2) this is most explicit in Lemmas 4-5 but we don’t quite think the question of technical novelty is applicable per se for an orthogonal ML paper studying a new estimand.
> >
> > (3) Great idea, we included an introductory paragraph at the beginning of the theoretical analysis section in lines 241-245.
> >
> >
> > Minor issue: thanks for pointing this out, fixed.

---

> > > ### Author Response · Authors · 2024-12-02
> > > **Open to further discussion**
> > >
> > > Dear Reviewer ynuS,
> > >
> > > Thank you for your thorough review and thoughtful feedback. We appreciate your recognition of our work's key contribution in avoiding dependence on unstable inverse behavior policy weights, which has been a significant barrier in applying orthogonal ML to offline RL.  We've addressed your concerns through three key improvements:
> > >
> > > 1. We have enhanced our methodological contributions by better explaining our innovations in orthogonality analysis and clearly distinguishing our approach from prior work, particularly Lewis and Syrgkanis 2021. This includes a more precise explanation of how our framework addresses the challenges unique to offline RL.
> > > 2. We've strengthened our technical presentation by providing a clearer exposition of the theoretical analysis, particularly focusing on policy optimization convergence and the crucial role of margin assumptions. The revised framework better demonstrates how our approach handles these complex aspects.
> > > 3. We've implemented important structural improvements, including a new introductory paragraph to the theoretical analysis section, enhanced presentation of Theorem 3, and various technical refinements that improve overall clarity and precision.
> > >
> > > Does our revised manuscript better reflect these improvements? Our aim is not merely to address your suggestions but to surpass your expectations, crafting a submission that resonates with a wider audience. We value your insights on any perspectives that could benefit from additional enhancement.
> > >
> > > Best,
> > >
> > > Authors

---

### Official Review · Reviewer_HbWM · 2024-11-01

**Soundness:** 4
**Presentation:** 3
**Contribution:** 2
**Rating:** 6
**Confidence:** 4

**Summary:**

The paper brings the tools from orthogonal ML into offline
reinforcement learning, with an orthogonalized estimator for the
"difference in Q function" in the offline policy
evaluation/optimization settings. The high level point is that the
difference of Q functions \tau(s_t) = Q^\pi(s_t,1) - Q^\pi(s_t,0) satisfies
the moment equation:

r_t + \gamma Q^\pi(s_{t+1},a_{t+1}) - E[r_t+\gamma
Q^\pi(s',\pi_b(s')|s_t] = (a_t - \pi_b(1|s_t))\tau(s_t) + \eps

This motivates a strategy where all unknown quantities above except
for \tau are estimated in the first stage (as nuisance parameters) and
then plug-in version of these are used to estimate \tau in the second
stage. This can be combined with dynamic programming for both policy
evaluation and policy optimization, resulting in the two main
algorithms in the paper.

As far as theoretical results, I think they can be summarized (and
simplified a bit) as follows: suppose all the nuisance functions can
be estimated at a n^{1/4} rate, then we can estimate \tau at a n^{1/2}
rate. For policy optimization, this becomes non-trivial because the
nuisance function at time $t$ is a function of the primary estimand
$\tau_{t+1}$, so one must show that this more complex nuisance
function can still be estimated at a fast rate.

Comments:

1. Overall, the paper is quite hard to read, even if you have a
background in both RL and orthogonal ML (as I do). I think the
notation could be improved significantly, and the theorem statements
are very difficult to parse/understand. Some simplifications would
help significantly I think, without compromising the main ideas of the
paper, for example:

- Work in the well-specified setting, so there is no need for \tau^n

- Don't worry about localized rademacher complexity

- Just do Massart margin condition

Basically, I get the impulse to consider the most general settings,
but I think this significantly compromises the readability of the
paper and isn't often very relevant toward the main points. One idea
is to write the main body of the paper in the simplest possible
setting that conveys the main technical and conceptual innovations and
then put the most general results in the appendix.

2. To this end, I actually couldn't figure out what is the technical
innovation here. I saw the paragraph saying that the policy
optimization part, specifically dealing with the dependence between
\hat{\tau}_{t+1} and the nuisance function for time t is the main
challenge. My first impression is that the margin assumption allows
one to turn a ell_2 guarantee into an ell_{\infty} guarantee, which
one can use to perhaps break this dependence? But honestly I didn't
really get it and the theorem statement is way too complicated for me
to try to infer what is happening.

3. I also think the paper could benefit from connecting to the prior
literature on offline RL theory. In particular how do requirements
like "bellman completeness" appear in the analysis. I don't think it
is possible to estimate Q functions, essentially at all, without such
assumptions in the offline RL setting, see e.g., [1]. So where does
such an assumption come into the analysis?

Overall, my main concern is that the paper is way too hard to read,
even for relative experts in the area. I think it will not be
appreciated by most of the ICLR audience. And I might be wrong about
this (due to difficulty in understanding the paper), but I currently
feel that in terms of technical novelty, the paper does not quite meet
the bar set by the conference.

References

[1] Dylan J. Foster, Akshay Krishnamurthy, David Simchi-Levi, Yunzong
Xu. Offline Reinforcement Learning: Fundamental Barriers for Value
Function Approximation. COLT 2022.

**Strengths:**

see above

**Weaknesses:**

see above

**Questions:**

see above

---

> ### Author Response · Authors · 2024-11-29
> **response to review and comments**
>
> Thanks for the review and the helpful comments. We have made major changes to the presentation in the updated revision and believe these should address your comments re: presentation.
>
> Please let us know if there is anything else we can clarify.
>
> Responses to comments:
>
> * 1. Yes, that is a great summary of the results (indeed we are presenting somewhat “standard” results in the realm of orthogonal ML). In lines 317-319 we had some discussion of the qualitative takeaways from the standard orthogonal estimation results but we realize this got buried in the presentation of the analysis. In the revision, we have overhauled the presentation along these lines by refactoring the product-error nuisance function estimate and focusing on presenting a simpler version of Theorem 3 which preserves the main qualitative takeaway. We also proceed under the well-specified setting for statements on rates in a new Assumption 6.
>
> * 2. “Margin assumption …  one can use to perhaps break this dependence” Ultimately this is what happens in the policy optimization result. At every timestep, we re-run the estimation approach to “peel off” estimation of the time-t difference-of-Q function. This prevents the use of the margin assumption from exponentiating convergence over multiple time-steps. On the other hand, we use an inductive argument to show that the dependence on estimated policy arises only in $Q$ functions and is higher-order via the margin assumption at every timestep. Ultimately we don’t pursue the tightest analysis because we ultimately seek to conclude that the error from policy-dependent nuisance functions is higher-order, rather than accumulating over the horizon. The analysis is sufficient to establish that our analysis translates to faster policy optimization, too.
>
> Our innovations are moreso conceptual and technical innovation via the new estimation framework. These issues of policy-dependent nuisance functions have been around for a while in dynamic treatment regimes (see discussion in Zhang et al; they ultimately use a heuristic localization). Unlike Zhang et al, which incurs dependence on the estimated policy via the definition of the population estimand (they optimize the DR policy value estimate itself), the estimated future policy only affects our convergence of the $Q$ function in our time-t loss function. And, unlike Zhang et al, we consider greedy policies with respect to a potentially rich difference-of-Q function, further avoiding issues of policy-dependent nuisance functions. We will add discussion on this point in the appendix.
>
> B. Zhang, A. A. Tsiatis, E. B. Laber, and M. Davidian. Robust estimation of optimal dynamic treatment regimes for sequential treatment decisions. Biometrika, 100(3):681-694, 2013.
>
> * 3. “I don't think it is possible to estimate Q functions, essentially at all, without such assumptions in the offline RL setting, see e.g., [1]. So where does such an assumption come into the analysis?”
>
>
> We certainly aren’t making any claims that it’s possible to estimate $Q$ functions without the structural assumptions that appear previously in the literature — the assumption comes into the analysis via the assumptions on product of nuisance function estimation rates. (In the revision we refactored this out into its own assumption, A5). If there is no structural condition like “bellman completeness”, then we expect $Q$ estimation to also fail, meaning A5 would not be satisfied. But because we only need the rate of convergence of $Q$ function, we are fairly agnostic about which version of Bellman completeness is (implicitly) implied via requiring consistent $Q$ estimation. Nor does estimation of a well-specified difference-of-Q function imply estimation of the policy value $E[V^\pi(s)]$ in general, so we certainly aren't circumventing any such lower bounds on policy value estimation.
>
> We added additional discussion in lines 306-311 in the revision, summarizing this response.
>
> Re other connections to offline RL theory – in lines 253-257 of the revision (also in the included submission) we discussed the connection to strong concentrability assumptions appearing in the offline RL literature. We do acknowledge that the offline RL literature has developed methods under weaker versions of this assumption; we view analogous improvements for our method as interesting directions for future work.
>
> Lastly, our overall approach using orthogonal ML generally implies that offline RL theory results improving convergence results for $Q$ estimation would apply directly via the black-box nuisance estimation assumption (A5).

---

> > ### Author Response · Authors · 2024-12-02
> > **Open to further discussion**
> >
> > Dear Reviewer HbWM,
> >
> > Thank you for reviewing our detailed revisions and responses. We've made substantial improvements to address your feedback, particularly:
> >
> > 1. Overhauled the presentation of our theoretical results, with clearer organization of assumptions and a simplified version of Theorem 3
> > 2. Added detailed explanations of our methodological innovations and their relationship to existing work
> > 3. Clarified how our approach connects to offline RL theory through the black-box nuisance estimation assumption
> >
> > Does our revised manuscript now better align with your expectations? We sincerely appreciate your detailed and thoughtful review, which has greatly contributed to improving our paper. With around 24 hours remaining, we remain fully open to further discussion to ensure our submission is as robust and polished as possible.
> >
> > Best,
> >
> > Authors

---

> > > ### Author Response · Authors · 2024-12-03
> > > **Kindly follow up**
> > >
> > > Dear Reviewer HbWM,
> > >
> > > Thank you again for reviewing our detailed revisions and responses! As the discussion is coming to a close, we have not had a chance to hear your comments on our refined version. However, we greatly value your feedback and hope the refined version meets your expectations. If you have any remaining questions or concerns about the manuscript, we welcome your input to help us ensure the final version is as strong as possible. We're pleased that we've been able to address most concerns through targeted revisions on specific sections. As we strive to meet all reviewers' expectations, please let us know if you have any questions or concerns.
> > >
> > > Best,
> > >
> > > Authors

---

### Official Review · Reviewer_zoM5 · 2024-11-05

**Soundness:** 3
**Presentation:** 1
**Contribution:** 3
**Rating:** 6
**Confidence:** 2

**Summary:**

The paper considers estimation of advantage function in offline RL setting. The authors use ideas from R learners, which was designed for heterogenous effect estimation, and propose an estimator for difference-in-Q function. By introducing three nuisances (Q, m, pi^b), the authors obtain an identifying conditional moments and relevant loss function, of which the proposed estimator is the empirical minimizer. The paper derives the estimation error, policy evaluation and policy optimization.

**Strengths:**

1. Incorporating ideas from R learner to estimating difference of Q function is interesting. The paper details the required error rate conditions for nuisances estimations for the proposed method to work, which is also technically challenging.

2. The results presented are extensive, including policy evaluation and optimization.

**Weaknesses:**

Some suggestions on presentation
1. Theorem 2 and 3: if the requirement on nuisances rates are the same, then it may be better to just state once.

2. Theorem 3 is long. It may be better to move some of them out as assumptions and explain.

3. More generally, the whole paper proceeds by introducing new definition and assumptions, and then stating algorithms / theorems. It may be better to add more explanations about assumptions and motivations.


Typos.
Line 133: u should be superscript, and why do we need X here?

**Questions:**

1. In theorem 3, is there a reason to expect the $\rho$ to increase?

2. What is the theoretical advantage of R-learning based method over traditional methods in terms of estimating (difference of ) Q function using offline data? Traditionally methods are based on Inverse Propensity Scoring, or its doubly robust variant. These methods can also be used to estimate difference in Q.

---

> ### Author Response · Authors · 2024-11-29
> **thanks for your review -- revision incorporates your suggestions**
>
> Thank you for your review and recognizing the significance of the contributions and extensive results! We have incorporated your suggestions for presentation into the revision and hope you find the revision satisfactory. In general these were all minor changes in exposition and presentation that did not change the substance of the analysis.
>
> Please let us know if you have any remaining questions for discussion or if we can clarify anything else.
>
> Weaknesses:
> * “If requirement on nuisances rates are the same”
> Thanks for pointing this out, indeed they rely on a high-level assumption on quantification of the product error nuisance rates. Thanks for the suggestion, we have refactored this out as an additional Assumption, explained the assumption concisely, and now Thm. 3 repeatedly invokes the assumption for different timesteps/policies.
> * Thm. 3 is long -- See response to all reviewers. We opted for presentation of the main high-level result in the main text via some simplifications/weaker rates.
> * Definition/assumption/theorem -- we have added additional expositions throughout after introducing these technical statements.
>
> Questions:
> 1. We don’t require them to increase, but this is related to the sequential nature of the estimation problems. If they are not increasing, it is not a problem, the final rate is just related to the slowest estimation error rate over all timesteps. Thanks for the feedback/your potential concern, to avoid such impressions, we have presented a version in the main text that in the end uses the slowest of all nuisance rates.
>
> 2. We’d like to first clarify that the “traditional methods” you mention based on “Inverse Propensity Scoring, or its doubly robust variant” have been used in general to estimate E[V^\pi(S_0)], i.e. the average policy value, and _not_ our estimand, $Q(s,a)-Q(s,a_0)$.
>
> This is a really crucial point that we want to be clear about, to the best of our knowledge *there are no prior methods that leverage double-robustness to for MSE convergence to estimate the entire difference of Q function* (except for the naive direct method which would simply difference two estimates of Q functions, again without double robustness). Please let us know if you had a specific prior paper in mind that does this.
>
> So, our R-learner based method develops a novel estimation approach and does just that. The theoretical advantage includes the speedup of statistical rates of convergence in mean-squared error estimation of $Q(s,a)-Q(s,a_0),$ rather than the averaged Q function that is the policy value, $E[V^\pi(S_0)]$.
>
> You do bring up an interesting point. One could ask, what is the closest version of traditional methods that estimates $Q(s,a)-Q(s,a_0)$? We do expect that recent approaches based on “pseudo-outcome regression” could be used to regress upon the doubly-robust score Jiang and Li 2016 / Thomas and Brunskill 2015, for example (or the naive IPW score). The R-learner based approach can be understood as a variance-weighted version of such an approach. To our knowledge however, this extension has not been fully developed and tested either. Even then, comparing our method to this hypothetical unpublished extension, our approach based on orthogonal estimation has the benefits of: 1) minimizing a variance-weighted version thereof and 2) the inverse propensity weights therefore do not appear directly in the loss function, reducing optimization issues which can be magnified when using stochastic-gradient based methods with importance sampling weights, as is common in more practical implementations of offline Q-learning.
>
> So overall, our approach is different from traditional such DR methods in OPE (though related), and we expect it also improves upon natural extensions of traditional DR estimation to $Q(s,a)-Q(s,a_0)$ (which again, have not appeared in the prior literature).

---

> ### Author Response · Authors · 2024-12-02
> **Open to urther discussion**
>
> Dear Reviewer zoM5,
>
> Thank you for recognizing our paper's technical soundness and contributions. We would like to ensure we have fully addressed your concerns in the revision. Specifically, we have focused on:
> 1. Improving presentation by restructuring Theorem 3, consolidating nuisance rates, and providing more detailed explanations for assumptions.
> 2. Clarifying the theoretical advantages of our R-learning method, particularly how it differs from traditional IPW/DR methods by estimating the entire difference of Q function rather than just policy value estimation.
>
> Have these revisions adequately addressed your concerns? As we still have time before the discussion period ends, we would be happy to clarify any of these aspects, especially regarding the technical content or the theoretical advantages of our approach.
>
> Best,
>
> Authors

---

> > ### Comment · Reviewer_zoM5 · 2024-12-02
> > **thank you!**
> >
> > The authors have improved presentation of the paper, and addressed my main concern about what traditional methods may or may not achieve in estimation of difference of Q. I increased my score.

---

> ### Author Response · Authors · 2024-12-02
> **Thanks for the supportive response and increasing the score!**
>
> Dear zoM5,
>
> Thank you for your valuable feedback and support again! We are delighted that our revisions have addressed your concerns with the **increased score**, particularly regarding the presentation and theoretical distinctions of our approach. Your insights have been instrumental in enhancing both the clarity and technical depth of our work.
>
> Best,
>
> Authors

---

### Official Review · Reviewer_WGH1 · 2024-11-09

**Soundness:** 3
**Presentation:** 2
**Contribution:** 3
**Rating:** 6
**Confidence:** 2

**Summary:**

This paper proposes an approach based on the concept of R-learner to estimate the difference of Q-function and subsequently policy optimization in reinforcement learning. The authors provide theoretical guarantees and conduct experiments on both synthetic data and CartPole to support their approach.

**Strengths:**

- The use of R-learner in causal inference to improve RL convergence rate appears novel.
- The author provides concrete theory on convergence guarantees. The technical analyses appear solid (Besides some typos, the proof seems sound).
- Experiments indicate that the proposed methods outperform baselines in terms of the mean square error in estimating differences of Q.

**Weaknesses:**

1. The main theoretical implications of this paper need to be explained more clearly. The authors mention they proved theoretical guarantees of *improved* convergence rate. It is unclear what works/results this paper is comparing with. Given the large RL literature, the author might need to discuss their improvements over existing approaches/results in much more clarity to position their contributions.

2. Assumption 1 (sequential unconfoundedness) needs justification. While it may be natural for other settings, whether this assumption is proper in RL is unclear. The assumption suggests actions only affect rewards but not the Markovian dynamic, which seems restrictive and also makes the problem potentially much easier.

3. My understanding is that the ultimate goal of the authors' approach is to obtain better RL policies, i.e., estimating difference of Q is only a means towards this ultimate goal. However, the experiments only consider the mean square error in estimating difference of Q. So there's a mismatch between the goal and the experimental results. Moreover, this limits the scope of baseline methods as most methods aim to obtain RL policies other than merely estimating difference of Q.

The above are my main concerns. In addition, the following might help improve the paper:
1. Fix typos, e.g., Eq. (3) $\epsilon(A_t)$ should be $\epsilon_t$, Eq. (4) $A$ should be $A_t$. Terminologies are not consistent, e.g., OrthDiff-Q v.s. $\tau$-TL v.s. DiffQ.
2. Experiments can be described more precisely. For example, in the Misaligned exo-endo experiment, $\beta_{dense}$ is not defined, and it is unclear how the data is generated.

**Questions:**

1. Does Assumption 1 hold in the numerical examples?
2. Connection with literature: E.g., how is Assumption 1 related to "Exogenous State MDP" in Dietterich et al., 2018? Regarding reward/value estimation, how is the proposed method compared to, e.g., Pan & Scholkopf, 2024, and others?
3. How should we understand or estimate $\alpha$ and $\delta_0$ in Assumption 5? This looks like an assumption for the sake of proof, and it's unclear how they can be understood for a given problem.
4. The cross-fitting part seems computationally intensive. What is the runtime of the experiment?

---

> ### Author Response · Authors · 2024-11-29
> **response and clarifications**
>
> Thanks for your review. We provide clarifications below.
>
> Weaknesses:
>
> * 1: Re other connections to offline RL theory and improvements to rates: our overall approach using orthogonal ML generally implies that offline RL theory results improving convergence results for $Q$ estimation would apply directly via the black-box nuisance estimation assumption (A5). Put simply, our results imply that just $n^{-¼}$ convergence for both $Q^{\pi^e}, \pi_b$ is needed to ensure $n^{-½}$ convergence of $Q(s,a’)-Q(s,a_0)$. Prior results would generally require $n^{-½}$ of $Q^\pi$ to achieve the same.
>
> * 2: We think there is a simple misunderstanding regarding Assumption 1. Importantly, it does not restrict the actions from affecting the state variable.
> The way we wrote it was confusing since we were trying to avoid introducing potential outcome notation, so the formal statement via conditional independence requires introducing potential outcomes S_{t+1}(A_t); while this is standard in causal inference, it is not standard in RL.
>
> We propose the best way forward instead is to revise Assumption 1 to the statement “Assumption 1: the underlying dynamics are from a Markov Decision Process on (S_t, A_t, R_t, …) and under the behavior policy, actions $A_t$ were taken with probability depending on observed states $S_t$ alone (and not on any unobserved states).”.
>
> To summarize, all that Sequential Unconfoundedness in Assumption 1 represents conceptually is that underlying dynamics are Markovian and that the behavior policy was not acting in a POMDP. It does not restrict actions from affecting transitions. It is much less restrictive than your initial impression, although we understand why you got that impression.
>
> We hope this improves your assessment of the paper, as what you perceived as a weakness was a simple misunderstanding we fixed in the updated revision. We hope that our fix introduces a clear conceptual statement. Importantly, we want to clarify that this was an oversight in our presentation, but what you thought this meant was not actually an assumption in the analysis or a restriction on the algorithm. It works for Markov decision processes.
>
> > In addition, the following might help improve the paper:
>
> * 1. Fixed, thanks.
> * 2. Fixed, thanks. We clarified that the "Misaligned exo-endo experiment" was just a small change from the prior DGP.

---

> ### Author Response · Authors · 2024-11-29
> **response to Questions**
>
> Responses to Questions:
>
> * 1. on A1: See our prior clarifications.
> Assumption 1 holds in the numerical examples by construction; future states $S_{t+1}$ and actions $A_t$ depend only on observed states $S_t$.
>
> * 2 We think our prior clarifications under Weaknesses should clear this up to. Assumption 1 is about the general suitability of MDPs for the underlying environment rather than POMDP, and is actually unrelated to the type of structural assumptions made in exogenous state MDP.
>
> The kinds of structural assumptions made in Dietterich 2018 relate to additional conditional independences defined among certain dimensions of the state variable, dimensions of the next state variable, and actions. These structural assumptions are more restrictive and pertain to substructures of the state variable. For example, stated as a conditional independence restriction between endogenous variables $s_{t+1}^\rho$ (a sub-partition or representation of the state) and exogenous variables $s_t^{\rho_c}$, the exogenous state MDP posits that $s_{t+1}^{\rho_c} \perp s_t^{\rho} \mid s_t^{\rho_c}, a_t$. The reward-filtered model instead posits that $s_{t+1}^\rho $ is endogenous and $s_t^{\rho_c}$ is exogenous if $s_{t+1}^\rho \perp s_t^{\rho_c} \mid s_t^\rho, a_t$.
>
> > Regarding reward/value estimation, how is the proposed method compared to, e.g., Pan & Scholkopf, 2024, and others?
>
> We mentioned this briefly in the related text but will inline the direct comparison. Pan & Scholkopf focused on ``direct” advantage estimation which elegantly avoids estimating future $Q$ functions; the drawback is that they require imposing nontraditional constraints on the advantage function, infinitely many nonconvex equality constraints, which is much harder than nonconvex unconstrained optimization. Their loss function is also just different.
>
> Overall the philosophies and goals are almost opposite and complementary: we precisely want to build on prior $Q$ function estimation and bring in additional information via the behavior policy to achieve faster rates of statistical convergence. We clarified in lines 86-88.
> We have updated in the revision a more concise version of this comparison.
>
> Regarding differences to value estimation: our method takes as black-box input estimates of $Q$ functions and is agnostic to which specific method is used; summarized in lines 191-195. Therefore, the convergence rates of prior methods relate to whether the product-error nuisance function rates (Assumption 5 in the revision, prev. in theorem statements) is satisfied.
>
> * 3. Re: $alpha, \delta_0$ in Assumption 5 (margin, now Assumption 7 in the revision):
>
> Let us explain more the role of $\alpha, \delta_0$: they quantify the strength of the assumption, i.e. how quickly the density falls away at the decision boundary. Importantly, they don’t need to be estimated in order to run the algorithm, they just quantify potential improvements in convergence rate. Typically an analyst would work with ``standard” structural assumptions on MDPs, such as linear etc; the margin assumption is common and other works have established $\alpha$ for common structural classes. In lines 332-334 of the revision, we described some values and clarify: for example, $\alpha=1$ for linear MDPs under a weak concentrability assumption, the margin assumption is always true if there is a nonzero gap between the value of optimal and sub-optimal actions.
>
> * 4. Cross-fitting only requires re-estimation of the nuisance functions, i.e. twice or three-times as many function estimations on half or a third of the data. Since the datasize also reduces the overall impact on time is that experiments would take _less_ than twice or three times as long (depending on whether cross-fitting for policy evaluation or optimization). The simple experiments run in a few minutes in parallel on a cluster with 24 parallel cores while the more complex experiments require ~20 minutes for a full comparison of all the methods.

---

> ### Author Response · Authors · 2024-12-02
> **Open to further discussion**
>
> Dear Reviewer WGH1,
>
> Thank you for your detailed review by mentioning our approach is novel and raising the insightful questions. We have worked to address the key points you raised, specifically:
> 1. We clarified that Assumption 1 is less restrictive than initially interpreted - it simply requires Markovian dynamics and that the behavior policy wasn't acting in a POMDP. We've revised its statement to be more precise and conceptually clear.
> 2. We explained how our approach connects to the literature:
>     * Compared to Pan & Scholkopf (2024), we take a complementary approach by building on Q-function estimation
>     * Our relationship to exogenous state MDPs (Dietterich 2018) involves different types of structural assumptions
>     * The margin conditions in our assumptions quantify convergence rate improvements for common structural classes
> 3. Regarding implementation, we've clarified that cross-fitting's computational impact is moderate, with experiments taking between a few minutes to ~20 minutes on a parallel computing setup.
>
> Combined with the updated draft with our point-wise response, have we sufficiently addressed your inquiries? We are open to further discussion if you have additional concerns or require further clarification.
>
> Best,
>
> Authors

---

### Author Response · Authors · 2024-11-30
**summary of revisions made and response to all reviewers**

We thank all the reviewers for their careful and thorough reviews, including helpful comments on presentation. We are glad the reviewers unilaterally recognized the importance of bringing orthogonal estimation to estimate the difference-of-Q function while, as ynuS points out, “avoiding the effects of unstable estimations in behavior policies, which is a common issue in offline RL”, which was indeed one of our key motivations. We also appreciate that the reviewers recognized the “concrete theory”(WGH1) and “solid/sound” analysis, with “extensive” results (zoM5)  including for the “non-trivial” (HbWM) case of policy optimization, and finally our focus in developing this method to “estimate more structured Q-function contrasts” which is “crucial” (ynuS).

All reviewers also pointed out that the paper is densely presented and the presentation can be improved for greatest clarity and succinctness. We greatly appreciate your thoughtful and helpful feedback and have taken this to heart, as well as your thought-provoking questions which helped us see further opportunities for improved presentation.

We are delighted to have updated a revised version of the paper, where edits are in *blue*. Major changes include simplified presentation of Theorems 2 and 3, adding more exposition in the Analysis section, refactoring out the product-rate error assumption. In individual responses to questions, we highlighted the changes we have made to address your concerns, reflected in minor changes throughout, including additional explanations of assumptions, lemmas, and theorems for improved flow. We have had to balance your nice suggestions for additional explanation with the space limit, deferring some additional exposition to the appendix when necessary. We believe this revision improves the presentation (keeping all of the substantive analysis and underlying results the same).

---

### Meta-Review · Area_Chair_fJYQ · 2024-12-21

**Metareview:**

This paper extends the R-learner framework from causal inference to offline reinforcement learning by introducing an orthogonalized estimator for Q-function differences. The proposed method is tailored for discrete-valued actions and adapts orthogonal machine learning techniques to estimate and optimize the difference between Q-functions under different actions, thereby advancing offline policy evaluation and optimization. The idea of leveraging the R-learner framework from orthogonal machine learning in offline RL is novel and encourages cross-disciplinary innovation in reinforcement learning theory.

However, after a thorough reading of the paper and consideration of the reviewer feedback, I am concerned that the paper is not in an accessible form, raising questions about its suitability for publication at ICLR. Of the four reviewers, two (zoM5, WGH1) recommend borderline acceptance but express low confidence (confidence level 2). Reviewer HbWM highlights that the paper is extremely difficult to read, even for experts in both RL and orthogonal ML, which hampers understanding of the paper’s technical novelty. Similarly, Reviewer ynuS notes that the technical challenges and contributions are not clearly articulated.

Given these concerns, the paper’s readability and accessibility emerge as critical issues. While the idea itself is promising and has potential, the current presentation significantly undermines its impact. I recommend rejection in its current form, with strong encouragement for the authors to focus on improving the clarity of their writing and explicitly addressing why the problem is technically challenging. With these improvements, the paper would be much better positioned for future submission. However, given the novelty of the idea and the borderline score, I would not be upset if this paper gets accepted.

---

SAC here. Given the borderline scores and the AC rejection, I took a look at the paper and the discussion here. I generally agree with the AC's decision that the paper needs much improvement on presentation. Below I will provide some further concrete suggestions.

In the rebuttal, the authors provided a clear story: their work uses $n^{-1/4}$ estimation of Q-function to achieve $n^{-1/2}$ estimation of the difference. Yet, nowhere in the main text can we find a clean and concise statement that formalizes this story into a mathematical statement. I strongly encourage the authors to reorganize the paper by putting such a theorem first and foremost and deferring the impenetrable Theorem 1 to later if not the appendix. A "cartoon" sketch of the main result could look like the following:

---

Assumption A. (Coverage condition, such as Assumption 4)

Assumption B. Assume we are provided with Q-function estimates with $n^{-1/4}$ rate.

Main Theorem. Under Assumptions 1 and 2 (and anything else that is essential), the estimation error of $\tau$ can be bounded as $\square n^{-1/2}$, where $\square$ is the problem-dependent constants.

---

The closest to this in the current manuscript is probably Theorem 2. Although the updated paper remarked below that this implies $n^{-1/2}$ rate, it is unclear how this follows from the mathematical statement of Theorem 2. Moreover, all problem-dependent constants ($\square$ above) are hidden. You assume bounded transition density in Assumption 4 and I expect the upper bound $c$ to appear in the theorem statement (currently it does not). In fact it's unclear to me how Assumption 4 translates to concentrability (the closest I've seen in the RL literature requires both upper- and lower-bounded density, whereas you only need an upper bound).

Another thing that can substantially help readability, as Reviewer HbWM pointed out, is to avoid superficial generalities. In addition to the ones already mentioned by HbWM (e.g., misspecification), another example is $\alpha$ which adds complications to Theorem 3, but I'd recommend getting rid of it (which I think corresponds to $\alpha=\infty$) since Assumption 7 is very contrived anyway. The probability in Assumption 7 is wrt the Lebesgue measure, which is a superficial measure that changes when the coordinates of the state representation are rescaled, unlike "invariant" measures like $d\_{\pi}$ which are much more fundamental to MDPs. Simply assuming that the gap assumption holds uniformly across the state space can help simply the presentation.

**Additional Comments On Reviewer Discussion:**

The reviewers primarily raised concerns regarding the interpretation of the results and their positioning within the context of existing technical challenges in orthogonal machine learning and offline reinforcement learning. While the authors' rebuttal addressed these concerns, it did not fully enhance the paper's overall readability.

---

### Decision · Program_Chairs · 2025-01-22

Reject